

# A weighted least squares approach to retrieve aerosol layer height over bright surfaces applied to GOME-2 measurements of the oxygen A band for forest fire cases over Europe

Swadhin Nanda[1,2], J. Pepijn Veefkind[1,2], Martin de Graaf[1], Maarten Sneep[1], Piet Stammes[1], Johan F. de Haan[1], Abram F. J. Sanders[3], Arnoud Apituley[1], Olaf Tuinder[1], and Pieternel F. Levelt[1,2]

[1]Royal Netherlands Meteorological Institute (KNMI), Utrechtseweg 297, 3731 GA De Bilt, The Netherlands
[2]Delft university of Technology (TU Delft), Mekelweg 2, 2628 CD Delft, The Netherlands
[3]University of Bremen, Institute of Environmental Physics, Otto-Hahn-Allee 1, 28359 Bremen, Germany

*Correspondence to:* Swadhin Nanda (nanda@knmi.nl)

**Abstract.** This paper presents a weighted least squares approach to retrieve aerosol layer height from top-of-atmosphere reflectance measurements in the oxygen A band (758 nm - 770 nm) over bright surfaces. A property of the measurement error covariance matrix is discussed, due to which photons traveling from the surface are given a higher preference over photons that scatter back from the aerosol layer. This is a potential source of biases in the estimation of aerosol properties over land,

which can be mitigated by revisiting the design of the measurement error covariance matrix. The alternative proposed in this paper, which we call the dynamic scaling method, introduces a scene-dependent and wavelength-dependent modification in the measurement signal-to-noise ratio in order to influence this matrix. This method is generally applicable to other retrieval algorithms using weighted least squares. To test this method, synthetic experiments are done in addition to application to GOME-2A and GOME-2B measurements of the oxygen A band over the August 2010 Russian wildfires, and the October

2017 Portugal wildfire plume over Western Europe.

## 1   Introduction

Algorithms that estimate properties of atmospheric species from satellite measurements of top-of-atmosphere (TOA) radiance (including spectral signatures of gases) in planetary atmospheres typically employ an inverse method based on least squares. In the case of retrieving terrestrial properties, this approach requires spectrally resolved measurements of the TOA Earth

radiance, solar irradiance, and a forward model as the minimal base ingredients to retrieve the state vector parameters (which are also model parameters). The goal of the least squares approach is to minimize a cost function, which aims to reduce discrepancies between the forward model and the measurement by iteratively manipulating state vector parameters. Upon minimization, the iterative scheme converges to a solution that, in principle, best describes the forward model's representation of the measurement.

Many atmospheric retrieval algorithms employ a weighted least-squares estimation (WLSE) method modified to include a-priori information on the state vector. An example of such an inverse method setup is optimal estimation (OE, Rodgers



(2000)), which is an attractive method particularly because of its efficacy in providing posteriori error statistics on the retrieved parameter. The KNMI aerosol layer height (ALH) retrieval algorithm uses an inverse method based on OE, and exploits the spectral structure of the near-infrared spectrum of the top-of-atmosphere radiance between 758 - 770 nm, where photons traveling through the Earth's atmosphere predominantly get absorbed by molecular oxygen. Oxygen is a well-mixed gas and

has a pressure-dependent spectral structure of its absorption lines (Min and Harrison, 2004); the further light in the oxygen A band passes through the atmosphere, the more it gets absorbed until it interacts with scattering species (such as clouds and aerosols) and scatters back to the TOA. It is this feature of the oxygen A band that has made it an attractive wavelength region for retrieving aerosol information (Gabella et al., 1999; Corradini and Cervino, 2006; Pelletier et al., 2008; Hollstein and Fischer, 2014; Sanghavi et al., 2012; Frankenberg et al., 2012; Wang et al., 2012; Sanders and de Haan, 2013; Sanders et al.,

2015; Sanders and de Haan, 2016). The algorithm is operational for the TROPOspheric Monitoring Instrument (TROPOMI) on board the Sentinel-5 Precursor (S5P) mission (Veefkind et al., 2012), and is also a part of the Sentinel-4 (S4) and Sentinel-5 (S5) missions (Ingmann et al., 2012) under the Copernicus satellite program of the European Union.

Due to the large spectral variability in absorption within the oxygen A band, the measured TOA radiance and the measurement noise have a high dynamic range. The minimization of the propagation of measurement noise to the final retrieval solution

should be a critical component of any retrieval algorithm. In WLSE, this is accomplished by the inverse measurement error covariance matrix which ranks the measurement on each detector pixel using the information available on the measurement noise. Due to the extent of the dynamic range of the measurement noise in the oxygen A band, this ranking matrix becomes a primary controlling entity; if the measurement noise is very large, the inverse noise variance is very low, which results in a lower rank to the measured signal from that specific detector pixel.

Since the measured signal is scene dependent, the spectral rank of each detector pixel is also scene dependent. This has special consequences over bright surfaces, where the dynamic range of the measured signal is much larger than over dark surfaces. Due to this, photons at wavelengths where the oxygen A band has a lower absorption cross section are less absorbed (subsequently traveling further into the atmosphere) and have a much larger representation in the WLSE method. A consequence of this, reported by Nanda et al. (2018), is that the retrieved ALH values are inaccurate for measurements over land.

In order to account for unknown instrument and model errors, Sanders et al. (2015) multiply the measurement error from L1b by two for their GOME-2 case studies and by ten in SCIAMACHY case studies (Sanders et al. 2018, manuscript in preparation) for retrieving ALH over ocean and land. They observe that increasing the measurement noise results in an increase in the number of retrieval convergences without significantly decreasing the accuracy of the retrieved ALH for the already-converged solutions. The method utilized by Sanders et al. (2015) does not change the shape of the noise spectrum since it is multiplied

by a constant. This paper investigates a vector-based weighing scheme (we call it the dynamic scaling method, as opposed to the formal approach which is unscaled OE), which dynamically varies from scene to scene; such a weighting scheme changes the shape of the noise spectrum itself. The objective of the dynamic scaling method is to influence the inverse measurement error covariance matrix in its choice in ranking the instrument's detector pixels in its spectral dimension in order to maximize sensitivity to aerosol layer height. The study discussed in this paper is a part of a series of papers discussing the ALH retrieval



algorithm developed at the KNMI (Sanders and de Haan, 2013; Sanders et al., 2015; Sanders and de Haan, 2016; Nanda et al., 2018).

The retrieval algorithm is described in section 2, which provides a description of the forward model and the formalism of OE. The incompatibility of retrieving aerosol properties from oxygen A band measurements with the formal design of the

measurement error covariance matrix are briefly discussed in the same section (section 2), before a full description of the proposed method in section 3 and a demonstration in a synthetic environment in section 4 are given. This method is applied to real data in section 5. The Russian wildfires in August 2010, which were discussed by Nanda et al. (2018), are revisited to compare the two approaches. The data are derived from the GOME-2A (Global Ozone Monitoring Experiment on board the MetOp-A platform of the EUropean Organization for the Exploitation of METeorological SATellites, or EUMETSAT)

instrument, and validated with a co-located CALIPSO (Cloud-Aerosol Lidar and Infrared Pathfinder Satellite Observation of the National Aeronautics and Space Administration, or NASA) overpass. The dynamic scaling method is further applied to the Portugal fires plume over Western Europe on the 17th of October, 2017, using data from the GOME-2B instrument, with validation from the ground-based EUropean METeorological services NETwork (EUMETNET, Alexander et al. (2016)) ceilometer network in the Netherlands and Germany, along with radiosonde measurements of the relative humidity profile and

the back trajectory of the aerosol plumes. This demonstration is followed by the conclusion in section 6.

## 2 The ALH retrieval algorithm

The algorithm is comprised of a forward model and an inverse method. The forward model uses a radiative transfer model described by de Haan et al. (1987) to calculate the top-of-atmosphere (TOA) Earth radiance ($I$) in the oxygen A band. This is done by propagating incoming solar irradiance ($E_0$) in the oxygen A band through the Earth's atmosphere, which is described

by an atmospheric model. Finally, this model is fitted to the measured spectrum to retrieve primary unknowns, Aerosol Optical Thickness (AOT) and ALH. For more details, the reader may refer to Sanders et al. (2015).

### 2.1 The forward model

The atmospheric model describes the interaction of photons with various components of the Earth's atmosphere that either absorb photons or scatter it in different directions. The oxygen absorption cross-sections are derived from the NASA Jet

Propulsion Laboratory database, and first-order line mixing and collision induced absorption between $O_2$-$O_2$ and $O_2$-$N_2$ are defined from Tran et al. (2006) and Tran and Hartmann (2008). The scattering species in the atmosphere include gases and molecules that follow Rayleigh scattering principles, aerosols, clouds and the surface. At present, the algorithm assumes cloud-free scenes, since the presence of clouds can result in large biases in the retrieved ALH (Sanders et al., 2015; Sanders and de Haan, 2016). Aerosols are modeled as a single layer with a fixed thickness of 50 hPa. ALH is defined as the mid pressure of

the aerosol layer, converted to a height above the ground. The aerosol layer has a constant aerosol extinction coefficient and a fixed aerosol single scattering albedo ($\omega$). Scattering by aerosols is described by a Henyey-Greenstein phase function (Henyey and Greenstein, 1941) with an anisotropy factor $g$ of 0.7. This choice is motivated by the model's simplicity in describing





scattering, which facilitates faster radiative transfer calculations than a more complex Mie scattering model. Currently, the surface is modeled as Lambertian.

The radiative transfer calculations are done line-by-line within the wavelength range of 758 nm - 770 nm, which requires a large computational effort for a single retrieval per iteration. In order to reduce computational time per iteration, polarization is ignored. This is a viable step, since the Rayleigh scattering cross section is very low in the near-infrared region. To that extent, rotational Raman scattering is also ignored in the forward model.

The solar irradiance and Earth radiance are convolved with an Instrument Spectral Response Function (ISRF) $f_{\mathrm{ISRF}}(\lambda - \lambda_i)$ to simulate a spectrum observed by a satellite instrument. The TOA Reflectance ($R$) is computed as

$$y_i = R(\lambda_i) = \frac{\pi}{\mu_0} \frac{\int f_{\mathrm{ISRF}}(\lambda - \lambda_i) I(\lambda) d\lambda}{\int f_{\mathrm{ISRF}}(\lambda - \lambda_i) E_0(\lambda) d\lambda} \tag{1}$$

where $\mu_0$ is the cosine of the solar zenith angle $\theta_0$, and the subscript $i$ is the index of the spectral channel. For a more in-depth description of the forward model, please refer to Sanders et al. (2015). All synthetic spectra presented in this paper are from a hypothetical instrument with a Gaussian ISRF and a spectral resolution (FWHM) of 0.11 nm oversampled by a factor 3. These specifications are very similar to the Sentinel-4 Ultraviolet Visible and Near infrared (UVN) instrument. The sensitivity analyses conducted in this paper may also be applicable to instruments with a lower spectral resolution. Further on in this paper, experiments are conducted with measured spectra from the GOME-2 A and B instruments, which have a lower spectral resolution than the S4 UVN instrument.

## 2.2 The formal ALH inverse method

OE is a maximum a-posteriori (MAP) estimator designed to find a solution for unknowns $\boldsymbol{x}$ in the classic inverse problem described in Equation 2 as,

$$\boldsymbol{y} = \boldsymbol{F}(\boldsymbol{x}, \boldsymbol{b}) + \boldsymbol{\epsilon}, \tag{2}$$

where $\boldsymbol{y}$ is the vector of measurements (in this case, reflectance in the oxygen A band as a function of spectral channel index), $\boldsymbol{F}(\boldsymbol{x}, \boldsymbol{b})$ is the aforementioned forward model with the state vector $\boldsymbol{x}$ and other model parameters $\boldsymbol{b}$, and $\boldsymbol{\epsilon}$ represents the measurement noise (at each spectral point). The OE method, being a MAP estimator, requires the knowledge of a priori errors in the estimation parameters. These errors are represented by the a priori error covariance matrix $\mathbf{S_a}$ and the measurement noise covariance matrix $\mathbf{S}_\epsilon$. Because measurement noise is considered uncorrelated, $\mathbf{S}_\epsilon$ is diagonal. $\mathbf{S_a}$ is also considered diagonal since the state vector elements are assumed to be uncorrelated. The inverse method propagates these errors into the a posteriori error covariance matrix $\hat{\mathbf{S}}$ following Equation 3,

$$\hat{\mathbf{S}} = \left( \mathbf{K}^T \mathbf{S}_\epsilon^{-1} \mathbf{K} + \mathbf{S_a}^{-1} \right)^{-1}, \tag{3}$$





with $\mathbf{K}$ as the Jacobian, or the matrix of partial derivatives of $\boldsymbol{F}(\boldsymbol{x}, \boldsymbol{b})$ with respect to the state vector parameters $\boldsymbol{x}$ at the retrieval solution. Since the forward model is non-linear, a Gauss-Newton method is employed to minimize the cost function (Equation 4) towards a zero gradient,

$$\chi^2 = [\boldsymbol{y} - \boldsymbol{F}(\boldsymbol{x}, \boldsymbol{b})]^T \mathbf{S}_\epsilon^{-1} [\boldsymbol{y} - \boldsymbol{F}(\boldsymbol{x}, \boldsymbol{b})] + (\boldsymbol{x} - \boldsymbol{x_a})^T \mathbf{S_a}^{-1} (\boldsymbol{x} - \boldsymbol{x_a}), \tag{4}$$

with $\boldsymbol{x_a}$ as the a-priori state vector. The update to the state vector $\boldsymbol{x}_{n+1}$ for iteration $n$ is provided in Equation 5,

$$\boldsymbol{x}_{n+1} = \boldsymbol{x_a} + (\mathbf{K}_n^T \mathbf{S}_\epsilon^{-1} \mathbf{K}_n + \mathbf{S_a}^{-1})^{-1} \mathbf{K}_n^T \mathbf{S}_\epsilon^{-1} [\boldsymbol{y} - \boldsymbol{F}(\boldsymbol{x}_n, \boldsymbol{b}) + \mathbf{K}_n(\boldsymbol{x}_n - \boldsymbol{x_a})], \tag{5}$$

where $\mathbf{K}_n$ is the Jacobian at the $n^{\text{th}}$ iteration and $\boldsymbol{x}_n$ is the state vector at the $n^{\text{th}}$ iteration. The retrieval is said to converge to a solution when the state vector update is lower than the expected precision. The matrix $\mathbf{S}_\epsilon$ plays a very important role in the WLSE framework by, essentially, ranking each spectral point based on the absolute measurement error in order to reduce the effect of measurement noise in the retrieved parameter. This is done by the $\mathbf{S}_\epsilon^{-1}$ matrix, which assigns a relatively higher value for spectral points with a lower noise covariance, and vice versa. The spectral points with a higher $\mathbf{S}_\epsilon^{-1}$ value essentially have an overall stronger influence in the WLSE. The design of this WLSE framework makes the retrieval solution intrinsically dependent on the quality of the $\mathbf{S}_\epsilon^{-1}$ matrix. This matrix will always rank higher those spectral points that represent photons less absorbed by oxygen, i.e. those which travel through the atmosphere more easily, as the relative error at these spectral points is low. Because aerosols are weak scatterers of light, a large fraction of photons pass through the aerosol layer and interact with the surface before returning to the detector.

A spectrometer's detector pixel (in the spectral dimension) that contains a higher concentration of oxygen absorption lines receives less number of photons, in comparison to spectral points that contain fewer or no absorption lines. As a result of this, the relative error at these spectral points is larger, resulting in a lower signal-to-noise ratio (SNR). The expression of noise in the $\mathbf{S}_\epsilon$ matrix at each spectral point is, hence, dependent on the average absorption line strength within a spectral point. When the surface becomes brighter (e.g. over land), the number of photons traveling from the surface to the detector increases heterogeneously, depending on many contributing factors such as oxygen absorption line strength, aerosol optical thickness, aerosol layer height, and other atmospheric properties. In principle, however, the increase in signal for detector pixels with low oxygen absorption cross section is much higher than the same for detector pixels with a high oxygen absorption cross section. This will be reflected in the $\mathbf{S}_\epsilon$ matrix, which will (for example) rank measurements in the continuum higher than the same in the deepest part of the absorption band.

If the information on ALH is derived from absorption by oxygen, this design of the $\mathbf{S}_\epsilon^{-1}$ matrix does not encourage an accurate ALH retrieval. From a WLSE standpoint, the consequences of an increase in the number of photons in the TOA reflectance that travel to the surface can be quite significant, some of which are reported in Figure 7 of Nanda et al. (2018). A possible avenue of improving the $\mathbf{S}_\epsilon^{-1}$ matrix involves its dynamical manipulation. The manipulation proposed in this paper has been termed as the dynamic scaling method. The next section elucidates this method, with a comparative analysis against the formal inverse method, henceforth called the formal approach, presented further on in this paper.





## 3   The dynamic scaling method

The dynamic scaling method identifies favorable spectral points for ALH retrieval by first identifying spectral points that are the least favorable. The noise is increased at these unfavorable points, while keeping the noise at the other points unchanged. These favorable and unfavorable spectral points are identified using a class of vectors known as modifying vectors (with the symbol $\mathcal{M}$, and length equal to the number of spectral points).

To identify the unfavorable spectral points at which the measurement noise is to be modified, a modifying vector $\boldsymbol{\mathcal{M}}_{\mathrm{A_s}/z_{\mathrm{aer}}}$ is proposed as,

$$\mathcal{M}_{\mathrm{A_s}/z_{\mathrm{aer}}}(\lambda_i) = \frac{K_{\mathrm{A_s}}(\lambda_i)}{K_{z_{\mathrm{aer}}}(\lambda_i)} \ [\mathrm{hPa}], \tag{6}$$

where $K_{\mathrm{A_s}}(\lambda_i)$ is the derivative of the TOA reflectance with respect to surface reflectance at the $i^{\mathrm{th}}$ index of the spectral point on the detector, and $K_{z_{\mathrm{aer}}}(\lambda_i)$ is the same for $z_{\mathrm{aer}}$. In principle, the ratio of $\boldsymbol{K}_{\mathrm{A_s}}$ and $\boldsymbol{K}_{z_{\mathrm{aer}}}$ is used as an identification tool since our primary retrieval parameter is $z_{\mathrm{aer}}$ whose information reduces as $\mathrm{A_s}$ increases. This opposing nature is discussed by Nanda et al. (2018) (Figure 3 and Figure 4 in their paper), where they show an anti-correlation in the sensitivity of $\tau$ and $z_{\mathrm{aer}}$ in the atmospheric path contribution and surface contribution to the TOA reflectance. A large value in $\mathcal{M}_{\mathrm{A_s}/z_{\mathrm{aer}}}(\lambda_i)$ represents spectral points in the measurement with more sensitivity to $\mathrm{A_s}$ than to $z_{\mathrm{aer}}$. The motivation for choosing derivatives as the means for modification is also partly motivated from the fact that they are scene-dependent parameters, which make each modification unique to the scene.

Spectral points with a $\mathcal{M}_{\mathrm{A_s}/z_{\mathrm{aer}}}(\lambda_i)$ higher than a specific threshold value should have a limited representation in the estimation — these are the unfavorable spectral points. We define this threshold as the modifying threshold ($\mathcal{T}$), which is the $20^{\mathrm{th}}$ percentile value of $\boldsymbol{\mathcal{M}}_{\mathrm{A_s}/z_{\mathrm{aer}}}$. The threshold value set in our method has been chosen in a way to avoid scaling the deeper parts of the R and P branches in the A band. The choice of thresholding remains configurable to the user of this method, based on their requirements — in our case we have chosen to use a static rule for deciding the value of $\mathcal{T}$, but this could also be made dynamic. An example of the shape of $\boldsymbol{\mathcal{M}}_{\mathrm{A_s}/z_{\mathrm{aer}}}$ is provided in Figure 1 (top row).

The reason for increasing the noise at specific unfavorable spectral points is to increase the value of $\mathbf{S}_\epsilon$ at these points. With a higher $\mathbf{S}_\epsilon$ value, the $\mathbf{S}_\epsilon^{-1}$ value will be lower, and hence that spectral point will have a lower weight in the estimation. In principle, this is equivalent to artificially increasing noise of measurements that contain less sensitivity to aerosol layer height. To do this, the modified SNR (denoted as $\mathrm{SNR}_{\mathcal{M}}$) is defined as,

$$\mathrm{SNR}_{\mathcal{M}}(\lambda_i) = \begin{cases} \mathrm{SNR}(\lambda_i), & \text{if } \mathcal{M}_{\mathrm{A_s}/z_{\mathrm{aer}}}(\lambda_i) < \mathcal{T} \\ \mathrm{SNR}(\lambda_i)/\mathcal{M}_{\mathrm{A_s}/\tau}(\lambda_i), & \text{otherwise} \end{cases} \tag{7}$$





where $\mathcal{M}_{A_s/\tau}(\lambda_i)$ (belonging to the class of modifying vectors) is defined as the ratio between the derivative of the TOA reflectance with respect to the surface ($K_{A_s}(\lambda_i)$) and the same with respect to aerosol optical thickness ($\tau$) at 760 nm ($K_\tau(\lambda_i)$),

$$\mathcal{M}_{A_s/\tau}(\lambda_i) = \frac{K_{A_s}(\lambda_i)}{K_\tau(\lambda_i)} \text{ [-].} \tag{8}$$

The choice of modifying the SNR based on $\mathcal{M}_{A_s/\tau}$ arises from the fact that the amount of contribution by the aerosol layer

to the TOA reflectance depends on its optical thickness. In such a case, we are interested in how much this contribution fares against the contribution from the surface. Information on both of these contributions can be inferred from the ratio of $K_{A_s}$ and $K_\tau$, which have comparatively similar shapes. If the measurement of a spectral pixel $i$ is more sensitive to $A_s$, $\mathcal{M}_{A_s/\tau}(\lambda_i)$ will be larger, and hence the noise at i will be increased, following Equation 7.

To run a retrieval using the dynamic scaling method, the derivatives of the reflectance with respect to $A_s$, $z_{aer}$ and $\tau$ at 760

nm are calculated first, followed by the modification of SNR according to Equation 7. The state vector parameters $\tau$ and $z_{aer}$ are then retrieved using spectrum $SNR_\mathcal{M}$. Users of this method may choose to scale the measurement error covariance matrix at each iteration, since the derivatives change at each iteration. Nevertheless, we have chosen to do it semi-statically since the measurement error covariance matrix is a static matrix throughout every iteration.

Examples of modifying vectors and $SNR_\mathcal{M}$ are provided in Figure 1 (bottom row), which shows the robustness of the method

in scaling the SNR for different surfaces. The spectra generated in the figure represents two scenes with identical atmospheric parameters, solar and satellite geometries, but different $A_s$. $\mathcal{M}_{A_s/z_{aer}}$, $\mathcal{T}$ and $\mathcal{M}_{A_s/\tau}$ for different surfaces are different — this is important, since over-scaling the SNR can force the retrieval to rank the measurements of photons traveling from the upper parts of the atmosphere higher, while ignore the same from the lower parts of the atmosphere. This is why the modifying vector $\mathcal{M}_{A_s/\tau}$ is chosen as a dynamically scene-dependent parameter (according to Equation 8), such that the scaling is large

when $A_s$ is large (Figure 1, mid row). In the next section, the dynamic scaling method is demonstrated and compared to the formal approach (which is the unscaled OE method) for synthetically generated spectra.

## 4  Sensitivity Analyses

To demonstrate the dynamic scaling method, synthetic spectra are generated for randomly varying values in $z_{aer}$, $\tau$, solar-satellite geometry ($\theta$, $\theta_0$ and $\phi - \phi_0$), and $A_s$, while keeping other parameters constant. Noise is not added to the synthetic

spectra. This method of randomly generating model parameters for generating synthetic spectra gives a broad picture of the method's behavior. Table 1 provides a brief overview of the input model parameters chosen for generating these spectra. An error is introduced in the forward model during retrieval, and the bias in $z_{aer}$ (defined as retrieved - true) is used to assess retrieval. The a priori $z_{aer}$ and $\tau$ are set at 825 hPa and true $\tau$, respectively. While there are many possible sources of errors, this paper presents two kinds of errors, a) error in the thickness of the aerosol layer, and b) error in the surface albedo database.

A reason for limiting the retrieval experiment scope to these two errors in the atmospheric part of the forward model is due to the fact that they are one of the more common contributors to retrieval biases. In real cases, aerosol layers may not be





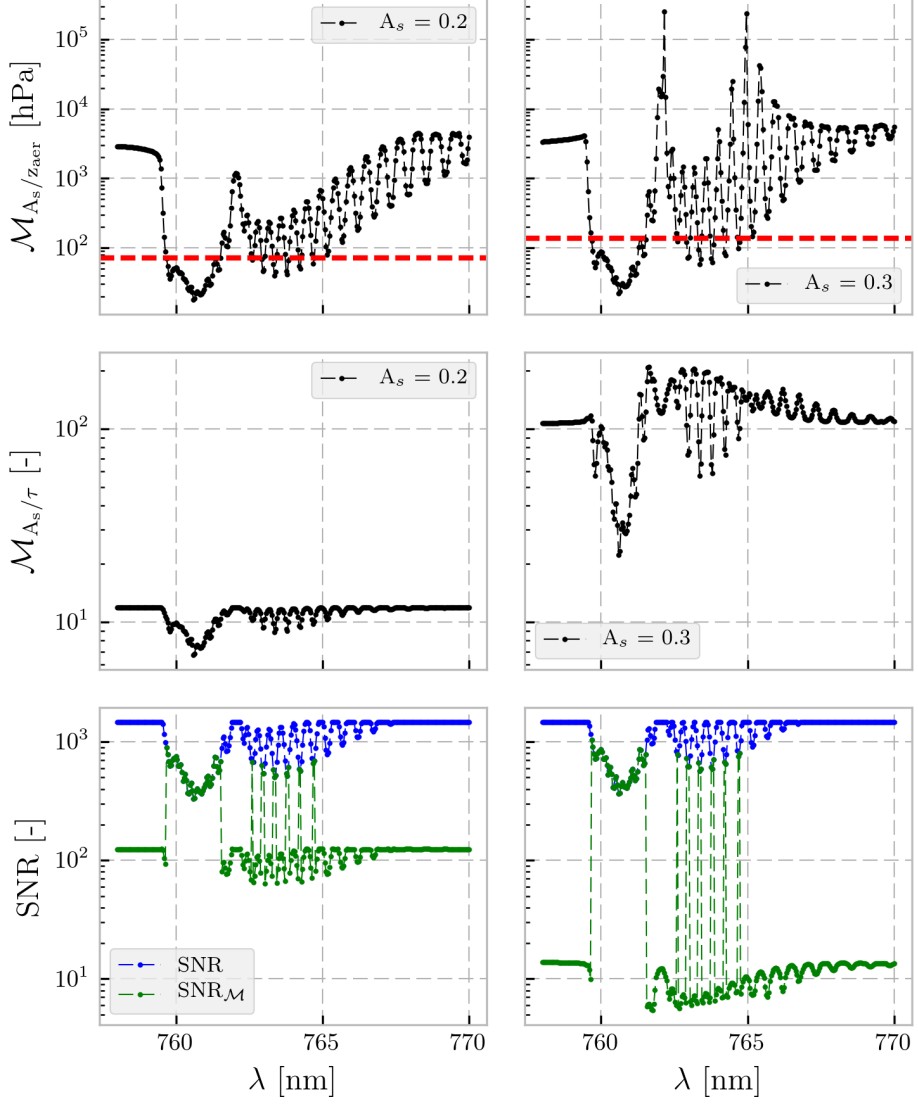

Figure 1. **Top row**: Modifying Vector $\mathcal{M}_{A_s/z_{aer}}$ as a function of wavelength $\lambda$. The solar zenith angle is $45°$, the viewing zenith angle is $20°$ and the relative azimuth angle is $0°$. The aerosol optical thickness ($\tau$) is 0.5 at 760 nm, over a surface with an albedo of 0.2 (left column) and 0.3 (right column) at 760 nm. The height of the aerosol layer is 900 hPa with a pressure thickness of 200 hPa. The aerosol single scattering albedo is 0.95 and the aerosol scattering is described by a Henyey-Greenstein phase function with an asymmetry factor of 0.7. The red dashed line represents the modification threshold value $\mathcal{T}$, which has been set at the $20^{\text{th}}$ percentile of $\mathcal{M}_{A_s/z_{aer}}$ in this example. **Middle row**: Modifying function $\mathcal{M}_{A_s/\tau}$, Equation 8 as a function of wavelength. **Bottom row**: The blue line represents the unscaled SNR whereas the green line represents the modified SNR according to Equation 7.



concentrated in a single layer of 50 hPa thickness, and the true surface albedo may vary significantly (to the order of 10% relative errors) from a monthly LER database depending on many parameters. In total, 2000 synthetic spectra are generated for each synthetic experiment and the parameters $z_{\mathrm{aer}}$ and $\tau$ are retrieved using both the formal approach and the dynamic scaling method, to be compared side-by-side. The results from analyzing biases in retrieved $z_{\mathrm{aer}}$ are plotted in Figure 2. Although

5  the dynamic scaling method is specifically designed for land, retrievals over surfaces with a low $A_{\mathrm{s}}$ (less than 0.1) are also included.

**Table 1.** Input parameters for synthetic experiments.

| name | value/remarks |
|---|---|
| **atmospheric parameters** | |
| $A_{\mathrm{s}}$ | 0.01 - 0.4 @ 760 nm (Lambertian) |
| $\tau$ | 1.0 - 5.0 @ 550 nm |
| $z_{\mathrm{aer}}$ | 600.0 - 900.0 hPa |
| $\omega$ | 0.95 |
| $g$ | 0.7 |
| Angstrom Exponent (Å) | 1.5 |
| temperature-pressure profile | mid-latitude summer |
| **instrument parameters** | |
| slit function FWHM | 0.11 nm |
| spectral oversampling factor | 3 |
| slit function shape | Gaussian |
| **solar-satellite geometry parameters** | |
| $\theta$ (viewing zenith angle) | 0° - 70° |
| $\theta_0$ | 0° - 70° |
| $\phi - \phi_0$ (relative azimuth angle) | $\phi = 180°$, $\phi_0$ varied between 0° - 360° |

### 4.1 Error in aerosol layer thickness

The synthetic spectra generated assume an aerosol layer thickness ($p_{\mathrm{thick}}$) of 100 hPa, whereas the retrieval forward model assumes a 50 hPa thickness. For simplicity, a PDF (denoted as $\varphi$) of the biases of retrieved $z_{\mathrm{aer}}$ is calculated, the peak of which represents the value of maximum frequency of occurrence, and the full-width at half maximum of which represents the spread.

10    In comparison with the formal approach (Figure 2a), the peak of $\varphi$ for the dynamic scaling method is closer to 0 hPa and has a larger magnitude (Table 2). The retrieval biases for $A_{\mathrm{s}} \leq 0.1$ and above 0.1 are indicative of the robustness of the dynamic scaling method in its scaling of the SNR (Table 2, $p_{\mathrm{thick}}$ bias row). For $A_{\mathrm{s}} \leq 0.1$, the retrieval biases from both dynamic scaling and formal approach are almost identical. The formal approach seems to retrieve 27 more pixels than the dynamic scaling method for $A_{\mathrm{s}} > 0.1$. An observation to note is that there are instances where even the dynamic scaling method can





result in large retrieval biases (Figure 2b). Generally however, the dynamic scaling method is shown to reduce retrieval biases in the presence of model errors in the aerosol layer thickness.

**Table 2.** Results of the retrieval accuracy of $z_{\text{aer}}$ from sensitivity analyses, split into two classes of $\text{A}_{\text{s}}$. The number of successful retrievals are reported in the 'retrieved' column. Columns with the heading A are the locations of the peak of $\varphi$, representing the $z_{\text{aer}}$ bias value with the highest frequency of occurrence. The same with B are the full width at half maximum of $\varphi$, representing the spread of $z_{\text{aer}}$ biases.

| Experiment | As | total spectra | Formal Approach | | | Dynamic scaling method | | |
| --- | --- | --- | --- | --- | --- | --- | --- | --- |
| | | | retrieved | A [hPa] | B [hPa] | Retrieved | A [hPa] | B [hPa] |
| $p_{\text{thick}}$ error | $\leq 0.1$ | 453 | 453 | 8.70 | 22.31 | 453 | 8.70 | 20.04 |
| | $> 0.1$ | 1547 | 1473 | 8.70 | 48.62 | 1446 | 3.34 | 38.76 |
| | | 2000 | 1926 | 8.70 | 44.18 | 1899 | 4.70 | 35.56 |
| $\text{A}_{\text{s}}$ error | $\leq 0.1$ | 451 | 451 | -2.00 | 17.84 | 451 | -2.00 | 14.36 |
| | $> 0.1$ | 1549 | 1335 | -2.00 | 178.27 | 1408 | -3.34 | 96.07 |
| | | 2000 | 1786 | -2.00 | 150.64 | 1859 | -3.34 | 81.85 |

## 4.2 Error in surface albedo database

For generating errors in surface albedo, randomly varying relative errors (with respect to the true surface albedo in the synthetic spectra) ranging between -10% to 10% were introduced to the retrieval forward model. The results heavily favor the dynamic scaling method, which shows a significant improvement in retrieval behavior over the formal method. The dynamic scaling method retrieves 73 more pixels than the formal approach (Table 2, $\text{A}_{\text{s}}$ error row), while also having a much smaller spread of retrieval biases around the peak (Figure 2c). For $\text{A}_{\text{s}} \leq 0.1$, the dynamic scaling method and the formal approach are almost identical, with the dynamic scaling method having a smaller spread. For $\text{A}_{\text{s}} > 0.1$, however, the dynamic scaling method improves the spread of the retrieval biases significantly. The mean biases for the dynamic scaling approach are slightly larger than the same for the formal approach, and the spread of retrieval biases in Figure 2d indicates that the dynamic scaling method does not necessarily improve retrieval biases for all cases. However, the method is able to improve both convergence and retrieval biases for a majority of the cases.

The analysis of retrieval biases from the synthetic sensitivity analyses are very encouraging for the dynamic scaling method. The method has shown significant improvements for $\text{A}_{\text{s}} > 0.1$ (at 760 nm) in the presence of two very relevant model errors. The fact that the dynamic scaling method is almost identical to the formal approach for $\text{A}_{\text{s}} \leq 0.1$ reaffirms the design of the modifying vector $\mathcal{M}_{\text{A}_{\text{s}}/\tau}$, which is intended to modify the SNR only if the modification is necessary. The success of the dynamic scaling method in a synthetic environment also confirms the fact that the design of the $\mathbf{S}_{\epsilon}^{-1}$ plays an important role in the biases of the retrieved $z_{\text{aer}}$. The next section applies the dynamic scaling method to measured spectra from GOME-2A and GOME-2B instruments over aerosol plumes from forest fire events in Europe.



## 5 Application to GOME-2 data

The GOME-2 instrument is a part of an operational mission by the European Organization for the Exploitation of Meteorological Satellites (EUMETSAT) to monitor trace gases and aerosols in the atmosphere. It is a spectrometer with an across-track scanning mirror that projects the TOA Earth radiance and solar irradiance through a prism on a grating to get information in the ultraviolet, visible and the near-infrared regions of the electromagnetic spectrum. In the oxygen A band, the spectral sampling interval is typically about 0.20 nm and the FWHM is 0.50 nm (Munro et al., 2016). The GOME-2 instrument is designed to have a footprint size of $80 \times 40$ km$^2$ in the oxygen A band. The instrument also measures the linear polarization of Earth radiance, which is important for correcting measured signal to calculate reflectance accurately.

In this section, measured spectra from the GOME-2A instrument on-board the Metop-A satellite over Russian wildfires on August 8, 2010 (Figure 3a) and the Portuguese fire plume with the GOME-2B instrument on-board the MetOp-B satellite on October 17, 2017 over Western Europe (Figure 3b) are used. The formal OE method is compared to the dynamic scaling method by using space-based and ground based validation data. The noise spectrum is derived from the GOME-2 Level 1-b product, which is a combination of the systematic and random error components of the measurements (EUMETSAT, 2014).

Auxiliary information required for these retrievals are meteorological data, surface albedo, and a-priori values for the optimal estimation (Table 4). The meteorological data required are temperature-pressure profiles and the surface pressure, derived from the ERA-Interim database from Dee et al. (2011). These meteorological parameters are available in regular space ($1° \times 1°$ spatial resolution) and time grids, and require interpolation to the satellite pixel's coordinates and time of record. This interpolation is done using nearest neighbor. The surface albedo database is derived from Tilstra et al. (2017) version 2.1, which has a resolution of $0.25° \times 0.25°$, derived from the GOME-2A instrument. The surface LER is chosen as the median of all LER database pixels intersecting the GOME-2 instrument pixel, at wavelengths 758 nm and 772 nm with linear interpolation used for calculating LER values at intermediate wavelengths. The algorithm assumes a cloud fraction of 0.0, and aerosols homogeneously distributed over the entire pixel. The a-priori aerosol optical thickness chosen is 0.8 at 760 nm, the aerosol layer top and bottom pressures are 775.0 hPa and 825.0 hPa, the aerosol single scattering albedo is 0.95 and the aerosol phase function is a Henyey-Greenstein model with an anisotropy factor of 0.7. The test cases chosen in this paper are relatively cloud-free, although not fully.

For validation, atmospheric lidar data from satellite and ground-based instruments are chosen. For the 2010 Russian wildfires, the lidar attenuated backscatter at 1064 nm from the CALIOP instrument (Cloud-Aerosol LIdar with Orthogonal Polarization) on board NASA's CALIPSO (Cloud-Aerosol Lidar and Infrared Pathfinder Satellite Observations) mission are used. These data have a very good representation of the scattering ability of clouds and aerosols in the atmosphere at a vertical resolution of 60 m and a horizontal resolution of 5 km. For the 2010 Russian wildfires, the CALIPSO overpass is at 10:45 UTC. All GOME-2A pixels co-located withing a 100 km vicinity of a CALIOP profile are considered for validation. For the October 17, 2017 Portugal fire plume over Western Europe, ground-based ceilometer data are used for validation (Table 3). These ceilometers are a part of the ALC (Automated Lidars and Ceilometers) network of the E-PROFILE observation program in the framework of the EUropean METeorological services NETwork (EUMETNET). The parameter used for validation is



the uncalibrated raw backscatter profile, since the paper focuses on qualitatively assessing the aerosol height retrievals with the lidar backscatter profiles. Lidar profiles within an hour of the satellite instrument overpass time are averaged into a single averaged profile, in order to reduce noise. These lidars have a vertical range of approximately 15 m, and record data at a very high temporal resolution, nominally every 6 seconds (Alexander et al., 2016). Although CALIOP data is available for the

5 plume over Western Europe for October 2017, CALIPSO does not have as good a co-location (both spatially and temporally) in comparison to the ceilometers.

**Table 3.** Ceilometer stations in Western Europe used for validating the the retrieved $z_{aer}$ from GOME-2B for plumes from the October 17, 2017 Portugal wildfires.

| name | institute | coordinates | GOME-2B overpass time |
|---|---|---|---|
| Hoogeveen | KNMI | 52.74° 6.59° | 09:31:10 UTC |
| Bonn | DWD | 50.74° 7.19° | 09:31:51 UTC |
| Luegde | DWD | 51.86° 9.27° | 09:31:18 UTC |
| Putbus | DWD | 54.36° 13.47° | 09:30:21 UTC |
| Luebeck | DWD | 53.81° 10.71° | 09:30:40 UTC |
| De Bilt | KNMI | 52.09° 5.17° | 09:31:21 UTC |
| Barth | DWD | 54.34° 12.71° | 09:30:25 UTC |
| Elpersbuettel | DWD | 54.06° 9.01° | 09:30:41 UTC |
| Soltau | DWD | 52.95° 9.80° | 09:30:56 UTC |
| Aachen | DWD | 50.79° 6.03° | 09:31:43 UTC |
| Hamburg | DWD | 53.65° 10.10° | 09:30:56 UTC |
| Braunschweig | DWD | 52.29° 10.44° | 09:31:05 UTC |

## 5.1 Russian wildfires on August 8, 2010

The wildfire plumes in and around Moscow on the 8[th] of August, 2010 are chosen as the test case for the dynamic scaling method. Anti-cyclonic conditions on this day meant that the region of interest was predominantly cloud-free. This case is the

10 same as analyzed in Nanda et al. (2018) (but with a smaller pixel selection to only focus on the plumes), with the exception that the study presented in the current paper uses a more-recent version of the surface LER product from Tilstra et al. (2017) with a larger amount of GOME-2A data incorporated into its creation. The inclusion of this more-recent LER database has slightly improved the results from the formal approach, but not significantly. A MODIS Terra image taken over the region on the same day (Figure 3a) shows that the plume, although thick, is non-homogeneously distributed in the scene, since the source of fires

are very close to the region of interest described in the test case. There are 85 GOME-2A pixels over the primary biomass burning plume that are considered for retrieving aerosol optical thickness and aerosol layer height. During the iterations, if the inverse method estimates non-physical state vector values (such as an aerosol layer below the surface and a negative aerosol



**Table 4.** Input data and algorithm setup for retrieving aerosol properties from GOME-2 measurements in the oxygen A band.

| parameter | source | remarks |
|---|---|---|
| radiance and irradiance | GOME-2A/GOME-2B | 3 minute granules |
| SNR measured spectrum | GOME-2A/GOME-2B operational Level-1b product | 3 minute granules |
| solar and satellite geometry | GOME-2A/GOME-2B Level 1-b data | 3 minute granules |
| surface albedo $A_s$ | Tilstra et al. (2017) GOME-2A LER at $0.25°$ x $0.25°$ grid at 758 nm and 772 nm | |
| temperature-pressure profile | ERA-Interim | nearest-neighbor interpolated |
| aerosol optical thickness $\tau$ | | state vector element, a-priori = 0.8 |
| aerosol layer height $h_{mid}$ [km] | | state vector element, a-priori = 800 hPa |
| aerosol single scattering albedo $\omega$ | | fixed at 0.95 |
| aerosol phase function $P(\theta)$ | | Henyey-Greenstein model with anisotropy factor $g$ of 0.7 |
| cloud mask | | none |
| validation (Russian wildfires in 2010) | CALIOP lidar profiles | 5 km $\times$ 5 km total attenuated backscatter at 1064 nm |
| validation (Portugal fires in 2017) | Alexander et al. (2016) | ground-based ceilometer network |

optical thickness or a cloud-like optical thickness) twice in a row, the retrieval is stopped and is said to have failed to converge. The algorithm also puts an upper cap of 12 iterations, beyond which the retrieval is also labeled to have failed to converge.

On applying the formal ALH retrieval approach, 49 pixels converge and 36 pixels do not converge to a solution (Figure 4 a,b). The retrieved aerosol optical thickness values are in excess of 6.0 in many cases — on average, the retrieved AOT is

5.34 with a standard deviation of 1.87 (Figure 5a, red). These values are not realistic, since aerosol optical thickness retrieved from the AErosol RObotic NETwork (AERONET) station in Moscow, which falls within one of the GOME-2A pixels, on the same day observed values between 1.0 at 870 nm and 1.5 at 675 nm between 09:00 UTC and 10:00 UTC. The distribution of retrieved $\tau$ appears to be spatially inconsistent with the aerosol plume observed by MODIS Terra (Figure 4, a). The formal approach misses the primary biomass burning aerosol plume. The average retrieved height of the plume is 0.5 km above the

ground, with a standard deviation of 0.15 km (Figure 5b, red histogram). Realistically, one can expect aerosols this close to the surface, especially if the boundary layer captures much of the pollution. However, aerosol-corrected boundary layer height modeled by Péré et al. (2014) for the same day over Moscow shows that the atmospheric boundary layer is approximately around 1.5-2.0 km altitude. Comparing the retrieval to co-located CALIPSO data in Figure 6 (blue markers), there are aerosols observed up to 4 km altitude, possibly in a multi-layered structure. Based on the CALIPSO observations and the modeled

height of the atmospheric boundary layer, the retrieved ALH seems to be biased low in the atmosphere, thus too close to the surface. These results are summarized in Table 5.





**Table 5.** Retrieval results from GOME-2 experiments. Columns marked with A, B, C and D are mean retrieved $z_{aer}$ (in km), standard deviation of retrieved $z_{aer}$ (in km), mean retrieved $\tau$ and standard deviation of the retrieved $\tau$, respectively. $n_{total}$ represents the total number of pixels in the scene, and $n_{ret}$ represents the number of retrieved pixels. $A_s$ avg represents the average surface albedo of the scene.

| case | $n_{total}$ | $A_s$ avg | formal approach | | | | | dynamic scaling method | | | | |
|---|---|---|---|---|---|---|---|---|---|---|---|---|
| | | | $n_{ret}$ | A [km] | B [km] | C [-] | D [-] | $n_{ret}$ | A [km] | B [km] | C [-] | D [-] |
| 2010 Russian wildfires | 85 | 0.19 | 49 | 0.5 | 0.15 | 5.34 | 1.87 | 78 | 1.37 | 0.367 | 4.82 | 2.04 |
| 2017 Portugal wildfires | 206 | 0.15 | 161 | 2.66 | 1.85 | 2.31 | 1.69 | 173 | 3.35 | 1.75 | 2.22 | 1.83 |

Applying the dynamic scaling method to the same scenario, we observe an increase in the number of convergences to 78 pixels out of the 85 chosen (60% increase compared to the formal approach), as shown in Figure 4 (c and d). The retrieved aerosol optical thickness is approximately 4.82, with a standard deviation of 2.04 (Figure 5a, blue histogram). While these retrieved AOT values are still unrealistic to the scene, the spatial distribution is consistent with the biomass burning plume seen by MODIS (Figure 4c). The retrieved aerosol layer height is, on average, 1.37 km, with a standard deviation of 0.367 km (Figure 5b, blue histogram). Looking at CALIPSO data, this value appears to be more realistic for the biomass burning plume (Figure 6, black markers), as the aerosol particles are located farther away from the surface.

### 5.2 Portugal fire plume over Western Europe on October 17, 2017

The October 2017 Portugal wildfires began in the third week of October. On the 16th of October, the hurricane Ophelia made landfall over Ireland as a mid-latitude cyclone. Due to the cyclonic conditions the forest fire aerosol plumes were pulled from Portugal into Western Europe along with Saharan desert dust (CAMS, 2017), which was observed the next day (Figure 3b). The aerosol plume from these fires are different from the aerosol plumes observed with the 2010 Russian wildfires case, primarily because the region of our interest is farther away from the fires; the plume over Western Europe appears to be more homogeneous. The GOME-2B overpass on the 17[th] October, 2017, is approximately around 09:30 UTC, and the MODIS image in Figure 3b is approximately around 11:00 UTC. Although some of these GOME-2B pixels may be cloud-contaminated, our retrieval assumes cloud-free conditions. This assumption can result in large values in retrieved aerosol heights and optical thicknesses. 206 GOME-2B pixels are chosen for this study. On average, the LER of this scene from the 2017 fires is 0.15 at 760 nm, whereas the same for the 2010 fires is 0.19.

Out of the 206 pixels, 161 pixels converge to a solution from the formal approach (Figure 7 a, b). The retrieved $\tau$ at 760 nm is on average 2.31, with a standard deviation of 1.69 (Figure 5c, red histogram). Typical retrieved $\tau$ over the plume seems to be around 3.0, which is too high of a value for this case since it disagrees with AERONET measurements, which show AOT values approximately between 2.0 and 1.0 at 675 nm and 870 nm over Lille during the GOME-2B overpass time. The retrieved $z_{aer}$ is, on average, approximately 2.66 km from the ground with a standard deviation of 1.85 km (Figure 5d, red histogram). Many of the pixels that do not converge seem to be cloudy (the bottom corner of the GOME-2B pixels, Figure 7a). The dynamic scaling method increases the number of convergences to 173 pixels (Figure 7 c, d). On average, this method retrieves an aerosol layer



height of 3.35 km, with a standard deviation of 1.75 km (Figure 5d, blue histogram). The average aerosol optical thickness at 760 nm retrieved is 2.22 with a standard deviation of 1.83 (Figure 5c, blue histogram).

Comparing the retrieved $z_{\mathrm{aer}}$ is to profiles from a ground-based ceilometer in De Bilt, Netherlands (Figure 8a, black profile), the first observation is that the dynamic scaling method seems to retrieve a height that is more representative of the top of

the aerosol layer, whereas the formal approach retrieves a more realistic aerosol height that is more-or-less at the centroid of the elevated layer's profile. It is, however, important to note that pulses from ceilometers are weak and tend to get attenuated beyond the bottom of the aerosol layer. Because of this, layers above these can appear as weak backscatterers even though they may not be. A radiosonde profile of the relative humidity reveals the presence of an atmospheric layer that extends well beyond the altitude range from where the lidar backscatter becomes progressively weaker. This profile also shows the presence of a

layer at the 200 - 400 hPa pressure levels, coinciding with a weak attenuated backscattered signal observed by the ceilometer in the same atmospheric level. A look into back trajectories, calculated using the TRAJKS model described in Stohl et al. (2001), shows that the pressure levels between 800 hPa to 600 hPa (at De Bilt) likely contains aerosols carried from Portugal to De Bilt (Figure 8b). The back trajectory of air mass at 250 hPa also passes through this peninsula, but may not contain biomass burning aerosols since the layer at this atmospheric level does not mix with the lower level (according to the TRAJKS calculations).

Following this, we have compared the retrieved $z_{\mathrm{aer}}$ from both methods to backscatter profiles from other ceilometer stations, reported in Figure 9. In general, while both the dynamic scaling method and the formal approach retrieve $z_{\mathrm{aer}}$ values that fall within the aerosol plumes, the dynamic scaling method retrieves heights that are slightly higher. This has to do with our conclusions from Figure 8.

The LER of a scene tells us which surface is brighter. In this case, the surface in the 2010 Russian fires was brighter

than the same in the 2017 Western Europe case. The values of the modifying vectors $\mathcal{M}_{\mathrm{A_s}/z_{\mathrm{aer}}}$ and $\mathcal{M}_{\mathrm{A_s}/\tau}$ over the two different scenes, however, can tell us the influence of the surface on the measurements itself, since these parameters are a direct comparison of the sensitivity of the measurement to aerosol properties and surface albedo. On average, $\mathcal{M}_{\mathrm{A_s}/z_{\mathrm{aer}}}$ and $\mathcal{M}_{\mathrm{A_s}/\tau}$ in the 2010 Russian wildfires case are much larger in comparison to the same for the 2017 Portugal fire plume over Western Europe (Figure 10). This suggests that backscatter from the surface for the 2010 Russian wildfires case plays a bigger role

in the measurements observed by the GOME-2 instrument. The dynamic scaling method is, hence, effectively able to apply a wavelength-dependent scaling of the SNR by relying on scene-dependent parameters. If the modifying vector $\mathcal{M}_{\mathrm{A_s}/\tau}$ is very low, aerosol properties retrieved from the dynamic scaling method will be approximately equal to the same from the formal approach. This is an example of the robustness of the method — the SNR should only be scaled when there is a need for it to be scaled.

**6 Conclusions**

Inversion algorithms that retrieve aerosol properties from spectral measurements in the oxygen A band (between 758 nm and 770 nm) can face a lot of trouble over land. This is primarily because of the location of oxygen A band band beyond the red-edge, a wavelength region with diminishing ability of vegetation to absorb solar radiation as wavelength increases. This



is especially the case when retrieving aerosol layer height using optimal estimation and radiative transfer models, as observed from Nanda et al. (2018), Sanders and de Haan (2016), and Sanders et al. (2015).

The optimal estimation framework, an application of the weighted least squares technique, is designed to rank data points (in this case, spectral points in the measured TOA radiance and solar irradiance) higher when the SNR is higher, in order to

reduce the influence of measurement error in the final retrieved solution. In the oxygen A band, these spectral points coincide with weak oxygen absorption cross sections, since low absorption equates to a high number of photons that can traverse through the atmospheric medium. Over oceans, due to its low albedo the number of photons that travel back from the surface are few. The signal recorded by satellites from an ocean scene, hence, predominantly arise from scattering and absorption by atmospheric species (in this case, aerosols). Over land, however, the number of photons that travel back from the surface

increases dramatically. Due to this, the optimal estimation framework ranks spectral points representing photons that have traveled back from the surface higher than the same from aerosol layers. This is the primary error source when it comes to biases in aerosol retrievals from oxygen A band measurements over land.

This paper introduces the dynamic scaling method, which is designed to retrieve aerosol layer height over bright surfaces from oxygen A band measurements. The core principle of this proposed improvement is the wavelength-dependent modifica-

tion of the measurement error covariance matrix by the subsequent wavelength-dependent modification of the signal-to-noise ratio of the measured spectrum, in order to reduce its preference towards photons that interact with the surface. The modification uses the scene-dependent Jacobian matrix, which makes it robust. The dynamic scaling method is compared with formal optimal estimation approach by retrieving aerosol layer height and aerosol optical thickness from synthetically generated spectra with randomly varied model parameters and model errors (that is, the forward models for simulation and retrieval have

different model parameters). The results from the synthetic experiments generally favor the dynamic scaling method, which shows a significant improvement of the accuracy of retrieved aerosol layer height in the presence of errors in the assumed aerosol geometric thickness and the surface albedo (up to 10% relative errors) in the model.

The dynamic scaling method is also demonstrated over real spectra by using GOME-2A and GOME-2B oxygen A band measurements of two separate wildfire incidences in Europe, one being the 8[th] of August, 2010 Russian wildfires and the other

being the more-recent 17[th] of October, 2017 Portugal wildfires. In the case of the 2010 Russian wildfires, the formal optimal estimation retrieval approach produces few convergences, and misses out the primary biomass burning aerosol plume (as observed from a MODIS Terra image). The retrieved aerosol optical thickness are unrealistically high and spatially inconsistent with the aerosol plume observed by MODIS Terra. Co-located CALIOP lidar profiles show that the retrieved aerosol layer height is biased low in the atmosphere, closer to the surface. The dynamic scaling method, on the other hand, improves the

number of converged pixels by 60% in comparison to the formal approach. The retrieved aerosol optical thickness is still not realistic, but the spatial distribution of the aerosol optical thickness, as compared to same observed in the MODIS Terra image, is consistent. The retrieved aerosol layer heights are also more realistic, as they are positioned close to the centroid of the CALIOP backscatter profile describing aerosols. For the Portugal wildfire plume in the 17[th] of October, 2017 over Western Europe, the dynamic scaling method does not increase the number of convergences significantly. The dynamic scaling

method retrieves aerosol layer heights that are only slightly higher, and aerosol optical thicknesses that are slightly lower in





comparison to the same from the formal approach. The retrieved heights from both method are compared to lidar profiles from the EUMETNET ACL network of ceilometers. The comparison shows that both methods retrieved heights that are within the profiles that could be associated with aerosol layers. Analyzing a radiosonde profile of the relative humidity and calculated back trajectories, it is observed that the ceilometer profiles miss higher aerosol layers due to attenuation of the signal at lower

atmospheric levels. This explains why the retrieved heights from the dynamic scaling method are slightly higher than the same from the formal approach.

In general, the dynamic scaling method improves the number of converged pixels. Between the two discussed cases, the dynamic scaling method provides a better improvement in the 2010 Russian wildfires case. This is primarily because the method is scene dependent. An important driver that determines the improvement of retrievals is the level to which the surface

influences the TOA reflectance, which is jointly influenced by two parameters — the surface albedo and the aerosol optical thickness. The average surface albedo of the scene for the 2010 Russian wildfires case was observed to be brighter than the same for the 2017 Portugal wildfires case. This is a possible explanation for the differences in the performance of the dynamic scaling method for the two cases.

The retrieved aerosol optical thickness is systematically lower for the dynamic scaling method in comparison to the formal

approach. A part of this can be attributed to the reduction of influence of spectral points in the measurement with a larger influence from the surface albedo. While this is expected, the method does not necessarily make the retrieved aerosol optical thickness more realistic. It may well be the influence of assumptions in aerosol properties such as aerosol single scattering albedo and the phase function. It could, however, also be that the method does not fully remove the influence of surface in the measured top-of-atmosphere reflectance signal. However, the goal of the aerosol layer height retrieval algorithm is to

estimate a diagnostic aerosol optical thickness rather than a realistic value. In this case, the dynamic scaling method improves the retrieved aerosol optical thickness's representativity of the aerosol plume over the 2010 Russian wildfires, and hence it's overall diagnostic quality.

The dynamic scaling method is designed to modify the signal-to-noise ratio to an extent that is necessary and sufficient in order to reduce the influence that photons traveling from the surface back to the detector have on the weighted least squares

estimate of aerosol properties. The choice of using the Jacobian to dictate the preference of weight least squares for spectral points in the measurement makes the dynamic scaling method a robust, generally-applicable retrieval setup. Results from this paper are applicable to other algorithms using weighted least squares techniques for retrieving atmospheric properties from measurements of top-of-atmosphere reflectance in the oxygen A band over bright surfaces.

*Competing interests.* The author declares no conflict of interests in the work expressed in this publication.

*Acknowledgements.* This research is partly funded by the European Space Agency (ESA) within the EU Copernicus programme under the project name 'Sentinel-4 Level-2 Processor Component Development', number AO/1-7845/14/NL/MP. We acknowledge EUMETSAT for





providing the GOME-2 L1b data. We thank Ina Mattis from the DWD and Marijn de Haij from the KNMI for providing us with valuable ceilometer profiles for validating satellite retrievals. We would also like to thank Marc Allaart from KNMI for providing the radiosonde profiles and Rinus Scheele from the KNMI for calculating the back trajectories.



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





**Figure 2.** Biases in retrieved $z_{aer}$ (in hPa) from synthetic measurements (2000 in total for each experiment) discussed in Section 4. The top row represents $z_{aer}$ biases in the presence of a model error in the thickness of the aerosol layer. The bottom row represents $z_{aer}$ biases in the presence of a model error in $A_s$. **(a), (c)** Probability distribution function $\varphi$ of retrieval biases. Blue line represents results from the dynamic scaling method, and the red line represents the same for the formal approach. **(b), (d)** 2D density plot showing the distribution of biases (density ranges from high in red to low in blue). The x axis represents biases from the dynamic scaling method, whereas the y axis represents biases from the formal approach.





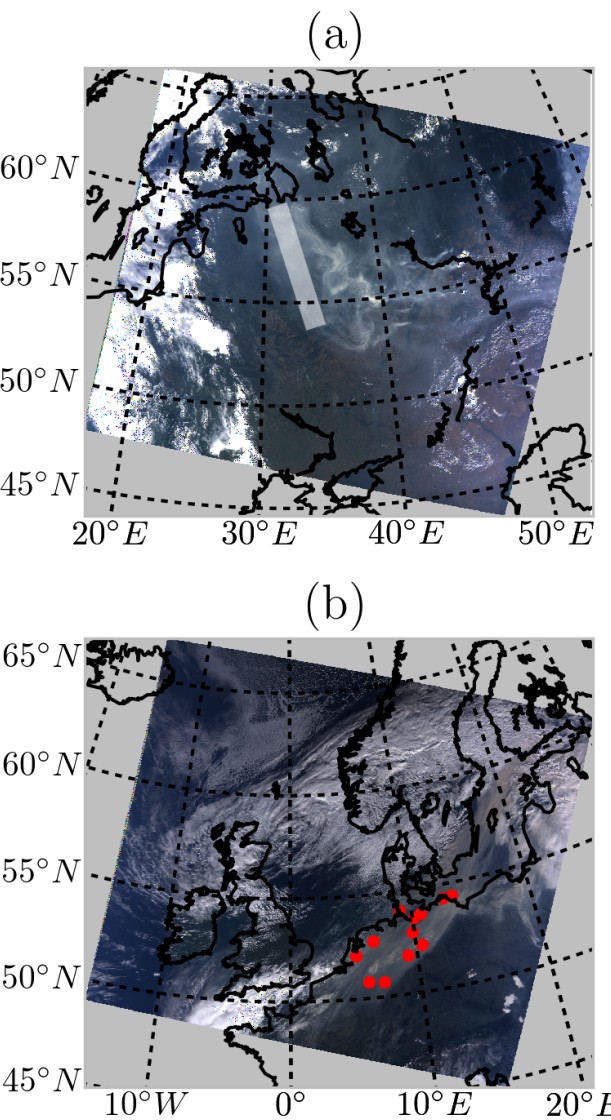

**Figure 3.** MODIS Terra images of the two test cases. **(a)** MODIS RGB composite on August 8, 2010 of the 2010 Russian wildfires. The white line represents an approximation of CALIPSO's ground track. **(b)** Portugal wildfire plume over Western Europe on October 17, 2017. Blue dots represent 12 ceilometer locations.



**Figure 4.** Results from processing 85 GOME-2A pixels over Russia on the $8^{\text{th}}$ of August, 2010 using the formal approach and the dynamic scaling method. Empty GOME-2A pixels with a white border represent non-convergences. **(a)** Retrieved $\tau$ at 760 nm from the formal approach. **(b)** Retrieved $z_{\text{aer}}$ from the formal approach. **(c)** Retrieved $\tau$ at 760 nm from the dynamic scaling method. **(d)** Retrieved $z_{\text{aer}}$ from the dynamic scaling method. The background image for all plots is a subset of the MODIS Terra image in Figure 3a.



**Figure 5.** Histograms of retrieved aerosol optical thickness ($\tau$, left column) and aerosol layer height ($z_{\mathrm{aer}}$, right column) from GOME-2A and GOME-2B pixels. Histograms in red are retrievals from the formal approach and the histograms in blue are results from the dynamic scaling method. **(a)** Retrieved $\tau$ from the GOME-2A pixels over the August 8, 2010 wildfires plume over Russian. **(b)** Retrieved $z_{\mathrm{aer}}$ from the GOME-2A pixels over the August 8, 2010 wildfires plume over Russian. **(c)** Retrieved $\tau$ from the GOME-2B pixels over the October 17, 2017 wildfires plume over Western Europe. **(d)** Retrieved $z_{\mathrm{aer}}$ from the GOME-2B pixels over the October 17, 2017 wildfires plume over Western Europe. The axes are adjusted for each plot.





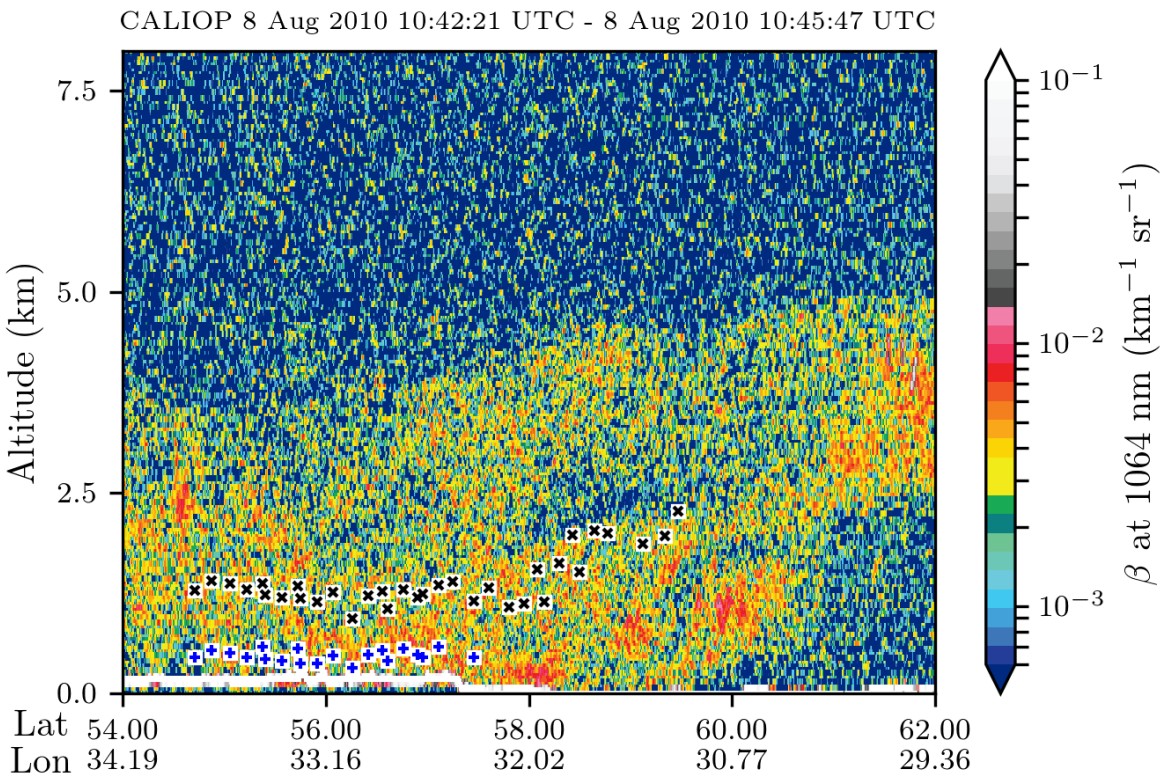

**Figure 6.** GOME-2A derived aerosol layer heights colocated within 100 km to the CALIPSO ground track (using great circle distance), plotted over attenated backscatter ($\beta$) of the CALIOP lidar at 1064 nm. The blue and black markers in white squares represent converged ALH from the formal approach and the dynamic scaling method, respectively.







**Figure 7.** Results from processing 206 GOME-2B pixels over Western Europe using the formal approach and the dynamic scaling method. Empty GOME-2B pixels with a white border represent non-convergences. **(a)** Retrieved $\tau$ at 760 nm from the formal approach. **(b)** Retrieved $z_{\mathrm{aer}}$ from the formal approach. **(c)** Retrieved $\tau$ at 760 nm from the dynamic scaling method. **(d)** Retrieved $z_{\mathrm{aer}}$ from the dynamic scaling method. The background image is a subset of the MODIS Terra image in Figure 3b.





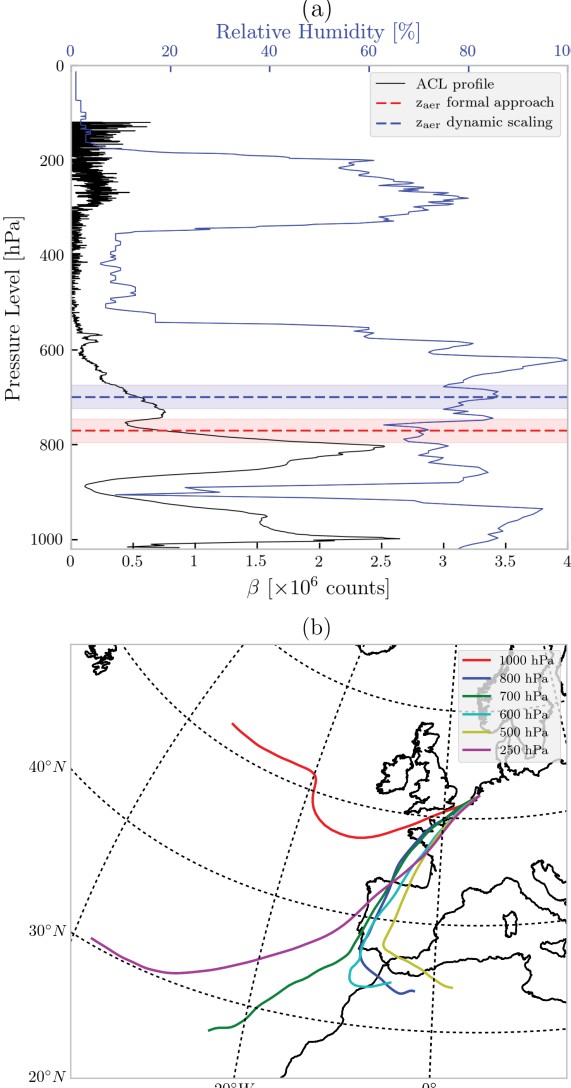

**Figure 8. (a)** Radiosonde profile of relative humidity (blue), plotted alongside an averaged raw attenuated backscatter profile (black) from the ceilometer at De Bilt, Netherlands. Both profiles are approximately around 13:00 UTC. The red and blue dashed line represents retrieved aerosol layer height using the formal approach and the dynamic scaling method, respectively. The red and blue shaded boxes represent the aerosol layer from the respective retrieval methods. The red and blue dashed line represents retrieved aerosol layer height using the formal approach and the dynamic scaling method, respectively. The red and blue shaded boxes represent the aerosol layer from the respective retrieval methods. **(b)** Back trajectories calculated for 17 October, 2017 at 13:00 UTC with the end point at De Bilt, and the sources going back to 3 days.







**Figure 9.** Validation of the retrieved aerosol layer height over Western Europe from ceilometers located in Netherlands and Germany from the CEILONET and DWD network. The black lines represent averaged ceilometer profiles of acquisitions 1 hour before and after the GOME-2B overpass over each location (600 profiles). The profiles are uncalibrated raw attenuated backscatter $\beta$ as a function of lidar range (in km). The gray shaded region represents the standard deviation of the profiles used to create the averaged profile. The red and blue dashed line represents retrieved aerosol layer height using the formal approach and the dynamic scaling method, respectively. The red and blue shaded boxes represent the aerosol layer from the respective retrieval methods.





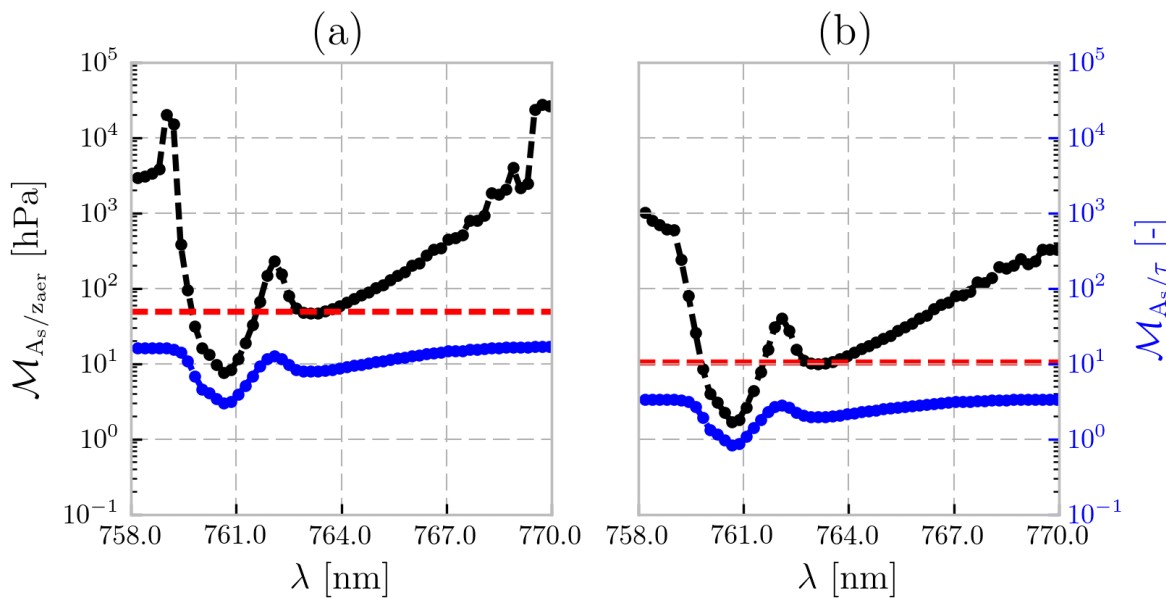

**Figure 10.** A comparison of the calculated matrices in the dynamic scaling method for all chosen GOME-2 pixels as a function of wavelength calculated for **(a)** the 2010 Russian wildfires, and **(b)** the 2017 Portugal wildfires. The black dotted line is the averaged modifying vector $\mathcal{M}_{A_s/z_{aer}}$ (Equation 6) and the blue line is the averaged modifying vector $\mathcal{M}_{A_s/\tau}$ (Equation 8) for all GOME-2 pixels chosen in each scene. The y-axis on the left is the range of values for $\mathcal{M}_{A_s/z_{aer}}$, and the same on the right is for $\mathcal{M}_{A_s/\tau}$. The red line is the averaged modifying threshold $\mathcal{T}$, which is set at the $20^{\text{th}}$ percentile of $\mathcal{M}_{A_s/z_{aer}}$.