# Peer review of "A weighted least squares approach to retrieve aerosol layer height over bright surfaces applied to GOME-2 measurements of the oxygen A band for forest fire cases over Europe"

_Atmospheric Measurement Techniques, 2018_

## Referee Comment (RC1) · Anonymous Referee #1 · 18 Apr 2018

**Review of the manuscript 'A weighted least squares approach to retrieve aerosol layer height over bright surfaces applied to GOME-2 measurements of the oxygen A band for forest fire cases over Europe' by Nanda et al.**

April 18, 2018

The manuscript describes a method to retrieve aerosol layer height by dynamically modifying the measurement error covariance matrix for a weighted least square approach. The manuscript is well-organised and include description and verification of the modified retrieval method. The manuscript is acceptable for publication after consideration of the suggestions for changes given below.

- **Page 4, line 6**: It is stated that Raman scattering is ignored. The impact of rotational Raman scattering in the $O_2$-A band has been quantified by Vasilkov et al. (2013). Please justify why you omit Raman scattering considering their results. Specifically, may Raman scattering significantly impact your dynamical scaling method?

- **Page 9, line 2**: Please define LER.

- **Page 9, line 14**: Please remove 'seems to'. The formal approach does not 'seems to' retrieve more pixels, it actually does so.

- **Page 10, line 13**: Please quantify 'majority of the cases', for example by giving the percentage.

- **Page 11, lines 22-24**: These numbers are provided in Table 4 and need not to be repeated here. If you choose to repeat them, include a reference to Table 4.

- **Page 11, line 32**: Table 3 is first referenced after Table 4. Please rearrange.

- **Page 13, line 5**: What is your reason for stating that the 'values are not realistic'? The Moscow station may not be representative as the plume is thick and non-homogeneous (manuscript page 12, line 14.) Have you compared the retrieved AOT values with MODIS AOT? That might shed light on how realistic the retrieved values are.

- **Page 13, line 6**: Please give the retrieved AOT values for the Moscow station pixel.

- **Page 14, line 4**: I would replace 'still unrealistic' with 'still high', but see also comment above for **Page 13, line 5**.

- **Page 14, line 18**: Please include reference to Table 5, that is, the sentence should end with: ' the 2010 fires is 0.19, see Table 5.'

- **Page 15, line 3**: Please change 'is to profiles from a' to 'to profiles from a'.

- **Page 16, line 23**: Please change 'demonstrated over real' to 'demonstrated for real'.

- **Page 16, line 29**: Please change 'improves' to 'increases'.

- **Page 16, lines 31**: Maybe change 'not realistic' to 'too high'?

- **Page 17, lines 19-22**: Your algorithm retrieves AOT and ALH. Here you state that the AOT is not necessarily a realistic value, but rather a diagnostic quality measure. If the AOT is of diagnostic quality only, how can then the ALH be a realistic value when they both come from the same retrieval? The discussion about AOT over these lines is rather unclear and maybe it is better to just leave it out.

- **Page 23, Fig. 3, caption**: Please include overpass times for the MODIS images.

- **Pages 24 and 27, Figs. 4 and 7**: Please change colour scale range in Figs. 4b,d and 7b,d so it agrees with the height ranges in Figs. 5b and 5d, respectively. As they are, Figs. 4b,d and 7b,d do not show the height structure.

- **Page 26, Fig. 6, caption**: Please change 'attenated' to 'attenuated'.

- **Page 28, Fig. 8, caption**: The following sentences are repeated twice: 'The red and blue dashed line represents retrieved aerosol layer height using the formal approach and the dynamic scaling method, respectively. The red and blue shaded boxes represent the aerosol layer from the respective retrieval methods.'

References

- Vasilkov, A., Joiner, J., and Spurr, R.: Note on rotational-Raman scattering in the O2 A- and B-bands, Atmos. Meas. Tech., 6, 981-990, https://doi.org/10.5194/amt-6-981-2013, 2013.

---

## Referee Comment (RC2) · Anonymous Referee #2 · 23 Apr 2018

The manuscript entitled "A weighted least squares approach to retrieve aerosol layer height over bright surfaces applied to GOME-2 measurements of the oxygen A band for forest fire cases over Europe" by Nanda et al. proposes the modified retrieval method from former's one. The method has been evaluated the effects of geometrical thickness of aerosol layer and surface albedo by using RT based simulated satellite signal. The authors have applied the method to large wildfire cases observed by GOME-2. Retrieved ALH and AOD are also compared with satellite- and ground- based lidar, and ground based sun photometric results. The proposed method improves the retrieved

area, and retrieved AOD is more reasonable compared to the formal method. From those results, the manuscript is suitable for the AMT journal, but a reviewer has some questions shown as below.

The authors tested the algorithm with synthetic experiments with high AOD (1<AOD<5) conditions. However the retrieved parameters of ALH are showed with low AOD (<1) results in Figure 5c. How much does the improved method increase the accuracy with low AOD (<1) case? What is the smallest value of AOD with the proposed method compared to formal one?

[Figure]

---

## Author Comment (AC1) · 7 May 2018

Response to RC1:

Thank you for your comments. Our response to your suggestions and questions as well as incorporated changes (in line with your suggestions) are detailed point-by-point in the following:

Page 4, line 6: It is stated that Raman scattering is ignored. The impact of rotational Raman scattering in the O2-A band has been quantified by Vasilkov et al. (2013).

Please justify why you omit Raman scattering considering their results. Specifically, may Raman scattering significantly impact your dynamical scaling method?

The primary reason for ignoring RRS is due to its computational requirement, which is significant. RRS is also dependent on the amount of Rayleigh scattering, which has a low cross section in the near-infrared. As such, RRS can be ignored for ALH retrievals.

Regarding the effect of RRS on the dynamic scaling method, we performed a synthetic experiment wherein spectra were generated with a 3-polynomial approximated RRS, and the retrieval forward model ignored RRS. We did this for 195 spectra, generated with randomly varying input parameters. The experiment excludes Ring spectrum and a differential ring spectrum, for simplicity.

The results are as follows.

The average bias in the retrieved aerosol layer height from the formal (unscaled) approach was approximately -7 hPa, whereas the same from the dynamic scaling approach was -11 hPa. The standard deviation of these biases were similar. So, ignoring RRS does affect dynamic scaling, just not to the degree that we can term significant, especially since including RRS will result in computational times doubling, sometimes tripling, the time from runs that ignore RRS. As such, ignoring RRS is a logical step.

Amendment to the manuscript:

Replaced 'To that extent, rotational Raman scattering is also ignored in the forward model.' with 'Because of the low Rayleigh Scattering cross section in the near-infrared, Rotational Raman Scattering can also be ignored.'

Page 9, line 2: Please define LER.

Accepted.

Amendment to the manuscript:

Replaced 'monthly LER database' with 'monthly database of Lambertian Equivalent
Reflectivity (LER) values'.

Page 9, line 14: Please remove 'seems to'. The formal approach does not 'seems to' retrieve more pixels, it actually does so. Accepted.

Amendment to the manuscript:

Replaced 'The formal approach seems to retrieve ...' with 'The formal approach retrieves ...'

Page 10, line 13: Please quantify 'majority of the cases', for example by giving the percentage.

Accepted. In order to comply, we calculated biases for retrievals with the formal approach, and compared their absolute value for the same with the dynamic scaling method. We found that, out of 2000 experiments, 1727 (or about 86%) of the retrievals had a lowered absolute bias value from using the dynamic scaling method.

Amendment to the manuscript:

Replaced 'However, the method is able to improve both convergence and retrieval biases for a majority of the cases.' with 'However, the dynamic scaling method improves convergence from 89.3% to 92.3%, and reduces bias for 86.4% of the cases.'

Page 11, lines 22-24: These numbers are provided in Table 4 and need not to be repeated here. If you choose to repeat them, include a reference to Table 4.

Accepted.

Amendment to the manuscript:

Replaced 'The algorithm assumes a cloud fraction of 0.0, and aerosols homogeneously distributed over the entire pixel. The a-priori aerosol optical thickness chosen is 0.8 at 760 nm, the aerosol layer top and bottom pressures are 775.0 hPa and 825.0 hPa, the aerosol single scattering albedo is 0.95 and the aerosol phase function is a Henyey-
Greenstein model with an anisotropy factor of 0.7.' with 'Algorithm settings are detailed in Table 3'.

Page 11, line 32: Table 3 is first referenced after Table 4. Please rearrange.

Accepted.

Amendment to the manuscript:

Replaced referencing of Tables 3 and 4: Table 4, containing algorithm settings, will be referenced before Table 3, which contains validation data location and GOME-2 colocation.

Page 13, line 5: What is your reason for stating that the 'values are not realistic'? The Moscow station may not be representative as the plume is thick and nonhomogeneous (manuscript page 12, line 14.) Have you compared the retrieved AOT values with MODIS AOT? That might shed light on how realistic the retrieved values are.

The retrieved AOT value using the dynamic scaling method over Moscow is 6.60 at 760 nm. With the formal approach, the retrieval does not converge. We did not check with the MODIS AOT value, and instead relied on the AERONET AOD values, which report AOD values of approximately 1.0 at 870 nm and 1.5 at 675 nm (meaning that AOT retrieved at 760 nm must lie in this range).

Amendment to the manuscript:

Please check our response to point 8.

Page 13, line 6: Please give the retrieved AOT values for the Moscow station pixel.

Accepted.

Amendment to the manuscript:

Replaced 'These values are not realistic, since aerosol optical thickness retrieved from the AErosol RObotic NETwork (AERONET) station in Moscow, which falls within one
of the GOME-2A pixels, on the same day observed values between 1.0 at 870 nm and 1.5 at 675 nm between 09:00 UTC and 10:00 UTC.' with 'These values are not realistic - the AErosol RObotic NETwork (AERONET) station in Moscow observed, on the same day, values between 1.0 at 870 nm and 1.5 at 675 nm between 09:00 UTC and 10:00 UTC, whereas our retrieval estimates an AOT of 6.60 at 760 nm over Moscow using dynamic scaling.'

Page 14, line 4: I would replace 'still unrealistic' with 'still high', but see also comment above for Page 13, line 5.

Accepted.

Amendment to the manuscript:

Replaced '... retrieved AOT values are still unrealistic to the scene, the spatial distribution is consistent with the biomass burning plume ...' with '... fitted AOT values are still high to the scene, the spatial distribution is consistent with the biomass burning plume ...'

Page 14, line 18: Please include reference to Table 5, that is, the sentence should end with: ' the 2010 fires is 0.19, see Table 5.'

Accepted.

Amendment to the manuscript:

Replaced ' On average, the LER of this scene from the 2017 fires is 0.15 at 760 nm, whereas the same for the 2010 fires is 0.19.' with ' On average, the LER of this scene from the 2017 fires is 0.15 at 760 nm, whereas the same for the 2010 fires is 0.19, see Table 5.'

Page 15, line 3: Please change 'is to profiles from a' to 'to profiles from a'.

Accepted.
Amendment to the manuscript:

Replaced '... is to profiles from a ground-based ceilometer in De Bilt ...' to '... to profiles from a ground-based ceilometer in De Bilt ...'.

Page 16, line 23: Please change 'demonstrated over real' to 'demonstrated for real'.

Accepted.

Amendment to the manuscript:

Replaced 'The dynamic scaling method is also demonstrated over real spectra by using GOME-2A and GOME-2B oxygen A band' with 'The dynamic scaling method is also demonstrated for real spectra by using GOME-2A and GOME-2B oxygen A band'

Page 16, line 29: Please change 'improves' to 'increases'.

Accepted.

Amendment to the manuscript:

Replaced 'The dynamic scaling method, on the other hand, improves the number of converged pixels by 60% in comparison to the formal approach' with 'The dynamic scaling method, on the other hand, increases the number of converged pixels by 60% in comparison to the formal approach'.

Page 16, lines 31: Maybe change 'not realistic' to 'too high'?

Accepted.

Amendment to the manuscript:

Replaced 'The retrieved aerosol optical thickness is still not realistic, but the spatial ...' with 'The fitted aerosol optical thickness is still too high, but the spatial ...'

Page 17, lines 19-22: Your algorithm retrieves AOT and ALH. Here you state that the AOT is not necessarily a realistic value, but rather a diagnostic quality measure. If

AMTD
the AOT is of diagnostic quality only, how can then the ALH be a realistic value when they both come from the same retrieval? The discussion about AOT over these lines is rather unclear and maybe it is better to just leave it out.

The question about the AOT being a diagnostic quantity compromising the realism behind ALH can be challenged by the fact that the retrieval of ALH depends more on the amount of light absorbed by oxygen, whereas the retrieval of AOT depends on the amount of light scattered back by aerosols. AOT does affect ALH, no doubt, and your concern regarding our discussion of AOT in these lines are well founded. To that extent, we accept to remove discussions of AOT in these lines.

Amendment to the manuscript:

Replaced 'However, the goal of the aerosol layer height retrieval algorithm is to estimate a diagnostic aerosol optical thickness rather than a realistic value. In this case, the dynamic scaling method improves the retrieved aerosol optical thickness's representativity of the aerosol plume over the 2010 Russian wildfires, and hence it's overall diagnostic quality.' with 'In any case, the dynamic scaling method improves the representativity of the fitted aerosol optical thickness of the MODIS Terra observed smoke plume.'

Page 23, Fig. 3, caption: Please include overpass times for the MODIS images.

Accepted.

Amendment to the manuscript:

Replaced '(a) MODIS RGB composite on August 8, 2010 of the 2010 Russian wildfires. The white line represents an approximation of CALIPSO's ground track. (b) Portugal wildfire plume over Western Europe on October 17, 2017. Blue dots represent 12 ceilometer locations.' with '(a) MODIS RGB composite on 08:50 UTC, August 8, 2010 of the 2010 Russian wildfires. The white line represents an approximation of CALIPSO's ground track. (b) Portugal wildfire plume over Western Europe observed AMTD
by MODIS Terra on 11:00 UTC, October 17, 2017. Blue dots represent 12 ceilometer locations.'

Pages 24 and 27, Figs. 4 and 7: Please change colour scale range in Figs. 4b,d and 7b,d so it agrees with the height ranges in Figs. 5b and 5d, respectively. As they are, Figs. 4b,d and 7b,d do not show the height structure.

Accepted. We also changed the color map of AOT to distinguish it from ALH.

Amendment to the manuscript: Replaced Figures 4 and 7 with images 1 and 2 included in this response.

Fig 1 (for Figure 4)

Fig 2 (for Figure 7)

Page 26, Fig. 6, caption: Please change 'attenated' to 'attenuated'.

Accepted.

Amendment to the manuscript:

Replaced '... ground track (using great circle distance), plotted over attenated backscatter ...' with '... ground track (using great circle distance), plotted over attenuated backscatter ...'

Page 28, Fig. 8, caption: The following sentences are repeated twice: 'The red and blue dashed line represents retrieved aerosol layer height using the formal approach and the dynamic scaling method, respectively. The red and blue shaded boxes represent the aerosol layer from the respective retrieval methods.'

Accepted.

Amendment to the manuscript:

Replaced 'The red and blue dashed line represents retrieved aerosol layer height using the formal approach and the dynamic scaling method, respectively. The red and blue
shaded boxes represent the aerosol layer from the respective retrieval methods.' with 'The red and blue dashed line represents retrieved aerosol layer height using the formal approach and the dynamic scaling method, respectively.'
Interactive

(a)

(b)

(c)

---

## Author Comment (AC2) · 7 May 2018

Thank you for your comments. Our response to your suggestions and questions as well as incorporated changes (in line with your suggestions) are detailed point-by-point in the following:

The authors tested the algorithm with synthetic experiments with high AOD (1<AOD<5) conditions. However the retrieved parameters of ALH are showed with low AOD (<1) results in Figure 5c. How much does the improved method increase the accuracy with

low AOD (<1) case? What is the smallest value of AOD with the proposed method compared to formal one?

To clarify, the AOT in the synthetic experiment are values at 550 nm, whereas the retrieved AOT in Figure 5c are at 760 nm. Our synthetic experiments have, so far, dealt with the issue of a bright surface hindering the accurate estimation of aerosol layer height. So, the real retrievals do include some aspects of the synthetic experiments.

If we split the AOD to two different classes, synthetic spectra for AOT <= 2.0 and AOT > 2.0 (AOT at 550 nm), we observe that, in general, the dynamic scaling method improves the accuracy of the retrieved aerosol layer height in the presence of a model error in the surface albedo. This improvement is much larger for scenes with AOT > 2.0.

The same split, when applied to synthetic experiments with a model error in aerosol layer pressure thickness shows that the dynamic scaling method better improves ALH retrieval accuracy for scenes containing optically thinner aerosol layers, in comparison to scenes with AOT > 2.0. This is because optically thin aerosol layers allow more influence of the surface in the ALH retrievals due to more photons interacting with the surface, whereas optically thick aerosol layer do not.

It is also important to note that the AOT is a fitted quantity, and is not the objective of the ALH retrieval algorithm. To this extent, we have now clarified throughout the document that AOT is fitted, and not retrieved.

For further clarity to the reader, we have added the range of AOT values in synthetic experiments in both 550 nm as well as 760 nm in Table 1.

These results are not included in a table format (as in Table 2) in the submitted manuscript. We have now added them in the text as follows.

Amendment to the manuscript:

Added to Page 9 lin 14: '... scaling and formal approach are almost identical. Splitting

the results to AOT <=2.0 and AOT > 2.0, it is observed that the dynamic scaling method reduces retrieval biases of ALH by 40% relative to the same from the formal approach for high aerosol loads, and about 11.5% for low aerosol loads. This is because a scene containing low aerosols allow for more interactions between photons and the surface, which results in ALH retrievals being biased closer to the surface. The dynamic scaling method ameliorates this behaviour by reducing the sensitivity of the retrieval algorithm to these photons.'

Added to Page 10 line 17: '... if the modification is necessary. A similar split of results for AOT <= 2.0 and AOT > 2.0 reveals that the dynamic scaling method is almost similar to the formal approach for low values of AOT, and only results in significant improvements if the scene contains sufficient aerosols. Relative to ALH biases from the formal approach, the biases from the dynamic scaling are reduced by 53% for AOT > 2.0, and is practically the same for AOT <= 2.0.'

Added entry to Table 1 under atmospheric parameters for the parameter 'tau' (or AOT) as follows:

Name: AOT value/remarks: 1.0 - 5.0 @ 550 nm (or 0.60 - 3.0 at 760 nm)

The following changes clarify that AOT is not a retrieval parameter, and is a fitted parameter:

Page 3 line 20-21:

'Finally, this model is fitted to the measured spectrum to retrieve primary unknowns, Aerosol Optical Thickness (AOT) and ALH' replaced with 'Finally, this model is fitted to the measured spectrum to retrieve the primary unknown ALH, while fitting the Aerosol Optical Thickness (AOT).'

Page 7 line 10 - 11:

'The state vector parameters $\tau$ and zaer are then retrieved using spectrum' replaced with 'The state vector parameters tau and zaer are then estimated using spectrum'

Page 9 line 3:

'Teach synthetic experiment and the parameters zaer and $\tau$ are retrieved using both the formal approach and the dynamic scaling' replaced with 'each synthetic experiment and the parameters zaer and tau are estimated using both the formal approach and the dynamic scaling'

Page 13 line 4:

'The retrieved aerosol optical thickness values are in excess of 6.0 in many cases — on average, the retrieved AOT is' replaced with 'The fitted aerosol optical thickness values are in excess of 6.0 in many cases — on average, the fitted AOT is' Page 13 line 8:

'The distribution of retrieved $\tau$ appears to be spatially' replaced with 'The distribution of fitted tau appears to be spatially'

Page 14 Table 5 Caption:

'mean retrieved $\tau$ and standard deviation of the retrieved tau' replaced with 'and the mean and standard deviation of the fitted tau'

Page 14 line 3-4:

'While these retrieved AOT values are still unrealistic to the scene' replaced with 'While these fitted AOT values are still unrealistic to the scene'

Page 14 line 16-17:

'large values in retrieved aerosol heights and optical thicknesses.' replaced with 'large values in retrieved aerosol heights and fitted optical thicknesses.'

Page 14 line 19:

'The retrieved tau at 760 nm' replaced with 'The fitted tau at 760 nm'

Page 14 line 20:

'Typical retrieved tau over the plume seems' replaced with 'Typical fitted tau over the plume seems'

Page 15 line 1-2:

'The average aerosol optical thickness at 760 nm retrieved is' replaced with 'The average aerosol optical thickness at 760 nm fitted is'

Page 16 line 27:

'The retrieved aerosol optical thickness are unrealistically high' replaced with 'The fitted aerosol optical thickness are unrealistically high'

Page 16 line 30:

'The retrieved aerosol optical thickness' replaced with 'The fitted aerosol optical thickness'

Page 17 line 14:

'The retrieved aerosol optical thickness is systematically' replaced with 'The fitted aerosol optical thickness is systematically'

Page 17 line 16:

'does not necessarily make the retrieved aerosol optical' replaced with 'does not necessarily make the fitted aerosol optical'

Page 17 line 21:

'the retrieved aerosol optical thickness's representativity' replaced with 'the fitted aerosol optical thickness's representativity'

Page 24 Figure 4 Caption:

'Retrieved tau at 760 nm from the formal' replaced with 'Fitted tau at 760 nm from the formal' 'Retrieved tau at 760 nm from the dynamic scaling method' replaced with 'Fitted

tau at 760 nm from the dynamic scaling method'

Page 25 Figure 4 Caption:

'Histograms of retrieved aerosol optical thickness replaced with 'Histograms of fitted aerosol optical thickness' 'Retrieved tau from the GOME-2A pixels over the August 8, 2010 wildfires' replaced with 'Fitted tau from the GOME-2A pixels over the August 8, 2010 wildfires' 'Retrieved tau from the GOME-2B pixels over the October 17' replaced with 'Fitted tau from the GOME-2B pixels over the October 17'

Page 24 Figure 4 Caption:

'Retrieved tau at 760 nm from the formal' replaced with 'Fitted tau at 760 nm from the formal' 'Retrieved tau at 760 nm from the dynamic scaling method' replaced with 'Fitted tau at 760 nm from the dynamic scaling method'

---

## Author Response (AR1)

Response to RC1:

Thank you for your comments. Our response to your suggestions and questions are detailed point-by-point in the following:

1.  Page 4, line 6: It is stated that Raman scattering is ignored. The impact of rotational Raman scattering in the O2-A band has been quantified by Vasilkov et al. (2013). Please justify why you omit Raman scattering considering their results. Specifically, may Raman scattering significantly impact your dynamical scaling method?

    *The primary reason for ignoring RRS is due to its computational requirement, which is significant. RRS is also dependent on the amount of Rayleigh scattering, which has a low cross section in the near-infrared. As such, RRS can be ignored for ALH retrievals.*

    *Regarding the effect of RRS on the dynamic scaling method, we performed a synthetic experiment wherein spectra were generated with a 3-polynomial approximated RRS, and the retrieval forward model ignored RRS. We did this for 195 spectra, generated with randomly varying input parameters. The experiment excludes Ring spectrum and a differential ring spectrum, for simplicity.*

    *The results are as follows.*

    *The average bias in the retrieved aerosol layer height from the formal (unscaled) approach was approximately -7 hPa, whereas the same from the dynamic scaling approach was -11 hPa. The standard deviation of these biases were similar. So, ignoring RRS does affect dynamic scaling, just not to the degree that we can term significant, especially since including RRS will result in computational times doubling, sometimes tripling, the time from runs that ignore RRS. As such, ignoring RRS is a logical step.*

    *Amendment to the manuscript:*

    *Replace '**To that extent, rotational Raman scattering is also ignored in the forward model.'** with 'Because of the low Rayleigh Scattering cross section in the near-infrared, Rotational Raman Scattering can also be ignored.'*

2.  Page 9, line 2: Please define LER.

    *Accepted.*

    *Amendment to the manuscript:*

    *Replace '**monthly LER database**' with 'monthly database of Lambertian Equivalent Reflectivity (LER) values'.*

3. Page 9, line 14: Please remove 'seems to'. The formal approach does not 'seems to' retrieve more pixels, it actually does so.
*Accepted.*

*Amendment to the manuscript:*

*Replace '**The formal approach seems to retrieve …**' with 'The formal approach retrieves …'*

4. Page 10, line 13: Please quantify 'majority of the cases', for example by giving the percentage.

*Accepted. In order to comply, we calculated biases for retrievals with the formal approach, and compared their absolute value for the same with the dynamic scaling method. We found that, out of 2000 experiments, 1727 (or about 86%) of the retrievals had a lowered absolute bias value from using the dynamic scaling method.*

*Amendment to the manuscript:*

*Replace '**However, the method is able to improve both convergence and retrieval biases for a majority of the cases.**' with 'However, the dynamic scaling method improves convergence from 89.3% to 92.3%, and reduces bias for 86.4% of the cases.'*

5. Page 11, lines 22-24: These numbers are provided in Table 4 and need not to be repeated here. If you choose to repeat them, include a reference to Table 4.

*Accepted.*

*Amendment to the manuscript:*

*Replace '**The algorithm assumes a cloud fraction of 0.0, and aerosols homogeneously distributed over the entire pixel. The a-priori aerosol optical thickness chosen is 0.8 at 760 nm, the aerosol layer top and bottom pressures are 775.0 hPa and 825.0 hPa, the aerosol single scattering albedo is 0.95 and the aerosol phase function is a Henyey-Greenstein model with an anisotropy factor of 0.7.**' with 'Algorithm settings are detailed in Table 3'.*

6. Page 11, line 32: Table 3 is first referenced after Table 4. Please rearrange.

*Accepted.*

*Amendment to the manuscript:*

*Replace referencing of Tables 3 and 4: Table 4, containing algorithm settings, will be referenced before Table 3, which contains validation data location and GOME-2 colocation.*

7.  Page 13, line 5: What is your reason for stating that the 'values are not realistic'? The Moscow station may not be representative as the plume is thick and nonhomogeneous (manuscript page 12, line 14.) Have you compared the retrieved AOT values with MODIS AOT? That might shed light on how realistic the retrieved values are.

*The retrieved AOT value using the dynamic scaling method over Moscow is 6.60 at 760 nm. With the formal approach, the retrieval does not converge. We did not check with the MODIS AOT value, and instead relied on the AERONET AOD values, which report AOD values of approximately 1.0 at 870 nm and 1.5 at 675 nm (meaning that AOT retrieved at 760 nm must lie in this range).*

*Amendment to the manuscript:*

*Please check our response to point 8.*

8.  Page 13, line 6: Please give the retrieved AOT values for the Moscow station pixel.

*Accepted.*

*Amendment to the manuscript:*

*Replace '**These values are not realistic, since aerosol optical thickness retrieved from the AErosol RObotic NETwork (AERONET) station in Moscow, which falls within one of the GOME-2A pixels, on the same day observed values between 1.0 at 870 nm and 1.5 at 675 nm between 09:00 UTC and 10:00 UTC.**' with 'These values are not realistic - aerosol optical thickness from the AErosol RObotic NETwork (AERONET) station in Moscow on the same day observed values between 1.0 at 870 nm and 1.5 at 675 nm between 09:00 UTC and 10:00 UTC, whereas our retrieval estimates an AOT of 6.60 at 760 nm over Moscow using dynamic scaling.'*

9.  Page 14, line 4: I would replace 'still unrealistic' with 'still high', but see also comment above for Page 13, line 5.

*Accepted.*

*Amendment to the manuscript:*

*Replace '**... retrieved AOT values are still unrealistic to the scene, the spatial distribution is consistent with the biomass burning plume ...**' with '... retrieved AOT values are still high to the scene, the spatial distribution is consistent with the biomass burning plume ...'*

10. Page 14, line 18: Please include reference to Table 5, that is, the sentence should end with: '
the 2010 fires is 0.19, see Table 5.'

*Accepted.*

*Amendment to the manuscript:*

*Replace '**On average, the LER of this scene from the 2017 fires is 0.15 at 760 nm, whereas
the same for the 2010 fires is 0.19.**' with ' On average, the LER of this scene from the 2017
fires is 0.15 at 760 nm, whereas the same for the 2010 fires is 0.19, see Table 5.'*

11. Page 15, line 3: Please change 'is to profiles from a' to 'to profiles from a'.

*Accepted.*

*Amendment to the manuscript:*

*Replace '**... is to profiles from a ground-based ceilometer in De Bilt ...**' to '... to profiles from
a ground-based ceilometer in De Bilt ...''.*

12. Page 16, line 23: Please change 'demonstrated over real' to 'demonstrated for real'.

*Accepted.*

*Amendment to the manuscript:*

*Replace '**The dynamic scaling method is also demonstrated over real spectra by using
GOME-2A and GOME-2B oxygen A band**' with 'The dynamic scaling method is also
demonstrated for real spectra by using GOME-2A and GOME-2B oxygen A band'*

13. Page 16, line 29: Please change 'improves' to 'increases'.

*Accepted.*

*Amendment to the manuscript:*

*Replace '**The dynamic scaling method, on the other hand, improves the number of
converged pixels by 60% in comparison to the formal approach**' with 'The dynamic scaling
method, on the other hand, increases the number of converged pixels by 60% in comparison to
the formal approach'.*

14. Page 16, lines 31: Maybe change 'not realistic' to 'too high'?

*Accepted.*

*Amendment to the manuscript:*

*Replace '**The retrieved aerosol optical thickness is still not realistic, but the spatial ...**' with 'The retrieved aerosol optical thickness is still too high, but the spatial ...'*

15. Page 17, lines 19-22: Your algorithm retrieves AOT and ALH. Here you state that the AOT is not necessarily a realistic value, but rather a diagnostic quality measure. If the AOT is of diagnostic quality only, how can then the ALH be a realistic value when they both come from the same retrieval? The discussion about AOT over these lines is rather unclear and maybe it is better to just leave it out.

*The question about the AOT being a diagnostic quantity compromising the realism behind ALH can be challenged by the fact that the retrieval of ALH depends more on the amount of light absorbed by oxygen, whereas the retrieval of AOT depends on the amount of light scattered back by aerosols. AOT does affect ALH, no doubt, and your concern regarding our discussion of AOT in these lines are well founded. To that extent, we accept to remove discussions of AOT in these lines.*

*Amendment to the manuscript:*

*Replace '**However, the goal of the aerosol layer height retrieval algorithm is to estimate a diagnostic aerosol optical thickness rather than a realistic value. In this case, the dynamic scaling method improves the retrieved aerosol optical thickness's representativity of the aerosol plume over the 2010 Russian wildfires, and hence it's overall diagnostic quality.**' with 'In any case, the dynamic scaling method improves the representativity of the retrieved aerosol optical thickness of the MODIS Terra observed smoke plume.'*

16. Page 23, Fig. 3, caption: Please include overpass times for the MODIS images.

*Accepted.*

*Amendment to the manuscript:*

*Replace '**(a) MODIS RGB composite on August 8, 2010 of the 2010 Russian wildfires. The white line represents an approximation of CALIPSO's ground track. (b) Portugal wildfire plume over Western Europe on October 17, 2017. Blue dots represent 12 ceilometer locations.**' with '(a) MODIS RGB composite on 08:50 UTC, August 8, 2010 of the 2010 Russian wildfires. The white line represents an approximation of CALIPSO's ground track. (b) Portugal*

*wildfire plume over Western Europe on 11:00 UTC, October 17, 2017. Blue dots represent 12 ceilometer locations.'*

17. Pages 24 and 27, Figs. 4 and 7: Please change colour scale range in Figs. 4b,d and 7b,d so it agrees with the height ranges in Figs. 5b and 5d, respectively. As they are, Figs. 4b,d and 7b,d do not show the height structure.

    *Accepted.*

    *Amendment to the manuscript:*
    *Will replace the Figure 4 and 7 with images 1 and 2 included in this response.*

    *Image 1 (to replace Figure 4):*

[Figure]

*Image 2 (To replace Figure 7):*

[Figure]

18. Page 26, Fig. 6, caption: Please change 'attenated' to 'attenuated'.

*Accepted.*

*Amendment to the manuscript:*

*Replace '... **ground track (using great circle distance), plotted over attenated backscatter …**' with '... ground track (using great circle distance), plotted over attenuated backscatter ...'*

19. Page 28, Fig. 8, caption: The following sentences are repeated twice: 'The red and blue dashed line represents retrieved aerosol layer height using the formal approach and the dynamic scaling method, respectively. The red and blue shaded boxes represent the aerosol layer from the respective retrieval methods.'

*Accepted.*

*Amendment to the manuscript:*

*Replace '**The red and blue dashed line represents retrieved aerosol layer height using the formal approach and the dynamic scaling method, respectively. The red and blue shaded boxes represent the aerosol layer from the respective retrieval methods.**' with 'The red and blue dashed line represents retrieved aerosol layer height using the formal approach and the dynamic scaling method, respectively.'*

Response to RC2:

Thank you for your comments. Our response to your suggestions and questions are detailed point-by-point in the following:

1. The authors tested the algorithm with synthetic experiments with high AOD (1<AOD<5) conditions. However the retrieved parameters of ALH are showed with low AOD (<1) results in Figure 5c. How much does the improved method increase the accuracy with low AOD (<1) case? What is the smallest value of AOD with the proposed method compared to formal one?

   *To clarify, the AOD in the synthetic experiment are values at 550 nm, whereas the retrieved AOD in Figure 5c are at 760 nm. Our synthetic experiments have, so far, dealt with the issue of a bright surface hindering the accurate estimation of aerosol layer height. So, the real retrievals do include some aspects of the synthetic experiments.*

   *If we split the AOD to two different classes, synthetic spectra for AOD <= 2.0 and AOD > 2.0, we observe that, in general, the dynamic scaling method improves the accuracy of the retrieved aerosol layer height in the presence of a model error in the surface albedo. This improvement is much larger for scenes with AOD > 2.0.*

   *The same split, when applied to synthetic experiments with a model error in aerosol layer pressure thickness shows that the dynamic scaling method better improves ALH retrieval accuracy for scenes containing optically thinner aerosol layers, in comparison to scenes with AOT > 2.0. This is because optically thin aerosol layers allow more influence of the surface in the ALH retrievals due to more photons interacting with the surface, whereas optically thick aerosol layer do not.*

   *These results are not included in a table format in the submitted manuscript. We propose to include them in the text as follows.*

   *Amendment to the manuscript:*

[revised manuscript text omitted]